# General heterostructure strategy of photothermal materials for scalable solar-heating hydrogen production without the consumption of artificial energy

Yaguang Li [1,2,3,6✉], Xianhua Bai[1,6], Dachao Yuan[1,2,6], Fengyu Zhang[1,4,6], Bo Li[1], Xingyuan San[1], Baolai Liang[1], Shufang Wang [1✉], Jun Luo [1,5✉] & Guangsheng Fu[1]

Solar-heating catalysis has the potential to realize zero artificial energy consumption, which is restricted by the low ambient solar heating temperatures of photothermal materials. Here, we propose the concept of using heterostructures of black photothermal materials (such as $Bi_2Te_3$) and infrared insulating materials (Cu) to elevate solar heating temperatures. Consequently, the heterostructure of $Bi_2Te_3$ and Cu ($Bi_2Te_3$/Cu) increases the 1 sun-heating temperature of $Bi_2Te_3$ from 93 °C to 317 °C by achieving the synergy of 89% solar absorption and 5% infrared radiation. This strategy is applicable for various black photothermal materials to raise the 1 sun-heating temperatures of $Ti_2O_3$, $Cu_2Se$, and $Cu_2S$ to 295 °C, 271 °C, and 248 °C, respectively. The $Bi_2Te_3$/Cu-based device is able to heat $CuO_x$/ZnO/$Al_2O_3$ nanosheets to 305 °C under 1 sun irradiation, and this system shows a 1 sun-driven hydrogen production rate of 310 mmol $g^{-1}$ $h^{-1}$ from methanol and water, at least 6 times greater than that of all solar-driven systems to date, with 30.1% solar-to-hydrogen efficiency and 20-day operating stability. Furthermore, this system is enlarged to 6 $m^2$ to generate 23.27 $m^3$/day of hydrogen under outdoor sunlight irradiation in the spring, revealing its potential for industrial manufacture.

[1] Hebei Key Lab of Optic-electronic Information and Materials, The College of Physics Science and Technology, Institute of Life Science and Green Development, Hebei University, 071002 Baoding, China. [2] College of Mechanical and Electrical Engineering, Key Laboratory Intelligent Equipment and New Energy Utilization of Livestock and Poultry Breeding, Hebei Agricultural University, 071001 Baoding, China. [3] State Key Laboratory of Photovoltaic Materials & Technology, Yingli Solar, 071051 Baoding, China. [4] Department of Materials Science and Engineering, China University of Petroleum Beijing, No. 18 Fuxue Rd., 102249 Beijing, China. [5] Institute for New Energy Materials & Low-Carbon Technologies and Tianjin Key Lab of Photoelectric Materials & Devices, School of Materials Science and Engineering, Tianjin University of Technology, 300384 Tianjin, China. [6] These authors contributed equally: Yaguang Li, Xianhua Bai, Dachao Yuan, Fengyu Zhang. ✉email: liyaguang@hbu.edu.cn; sfwang@hbu.edu.cn; jluo@tjut.edu.cn

Promoting industrial catalysis consumes a large amount of artificial energy, such as electricity[1] and fossil-derived energy[2], which need artificial conversion and input for their final usage. Therefore, constructing artificial-energy-input-free catalysis is the key to human sustainable development. Sunlight-driven catalysis is a typical type of artificial-energy-input-free catalysis that has the potential to solve the energy bottleneck of catalysis[3–5]. At present, sunlight-driven catalysis mainly includes photocatalysis via sunlight-photogenerated carrier-chemical paths and photothermal catalysis via sunlight-hot carrier-chemical paths[6,7], exhibiting great potential in many fields[8,9]. In addition to the above two kinds of sunlight-driven catalysis, using sunlight-converted thermal energy to drive thermal catalysis, that is, solar-heating catalysis, has aroused much attention[10,11]. In solar-heating catalysis, the photothermal material is the key for diverse solar-heating applications[12], and it is challenged by achieving a high solar-heating temperature upon irradiation by diluted ambient sunlight[13–17]. In previous reports[18], coordinated regulation of high solar absorption[19] and low thermal conductivity by materials such as black photothermal materials with porous[20], amorphous[21], and layered structures[22] was the main strategy used to improve solar-heating temperatures[23,24]. For example, a nanohybrid combining zeolitic imidazolate frameworks (ZIFs) and graphene was demonstrated to synergistically intensify sunlight absorbance (~98% solar-to-thermal conversion efficiency) and thermal energy insulation capability (ultralow thermal conductivity of ~0.2 W mK$^{-1}$) to achieve a reported maximum solar-heating temperature of 120 °C under 1 sun illumination (equal to an energy density of 1 kW m$^{-2}$)[25], which is still too low to drive most thermocatalytic reactions. From the ideal artificial-energy-input-free catalysis design point of view, it is necessary to propose a concept to further improve the solar-heating temperatures.

Hydrogen energy has been considered one of the foundations for future energy systems[26–28]. Owing to the storage limitations of hydrogen, such as high pressure, leakage, and extensive safety precautions, Olah proposed a methanol economy, as methanol can act as a hydrogen carrier in future hydrogen energy systems[29], which has the merits of high hydrogen storage density (99 kg m$^{-3}$), high degree of safety, low cost and compatibility with existing fossil energy systems[30–32]. However, hydrogen generation from methanol and water by methanol steam reforming ($CH_3OH + H_2O \rightarrow CO_2 + 3H_2$, MSR) requires a large-scale external energy input (16.47 kJ energy for 1 mol of $H_2$)[33]. A large amount of energy consumption has become the bottleneck for the large-scale application of methanol-hydrogen energy systems. Using sunlight to drive MSR is an attractive way to solve the artificial energy consumption problem[34–36]. As far as we know, the state-of-the-art sunlight-driven hydrogen production rate from methanol and water is ~46 mmol g$^{-1}$ h$^{-1}$[37–41], far behind industry requirements[42,43]. Therefore, development of an artificial-energy-input-free MSR mode with a greatly improved reaction rate is urgent for its applicability.

Herein, taking a typical narrow-band gap photothermal material, $Bi_2Te_3$, as an example, we demonstrate a concept to increase the sunlight irradiation temperature of photothermal materials in which a heterogeneous $Bi_2Te_3$ thin-film structure is synthesized on a Cu support ($Bi_2Te_3$/Cu) to simultaneously balance sunlight absorption and thermal radiation[22,44]. Sunlight absorption and thermal dissipation can be controlled by tuning the thickness of the $Bi_2Te_3$ thin film, resulting in a 1 sun heating temperature of 317 °C, which is much higher than that of pure $Bi_2Te_3$ (93 °C). Moreover, this strategy can also generally raise the 1 sun-heating temperatures of other photothermal materials to above 250 °C, and a reaction device based on $Bi_2Te_3$/Cu can heat catalysts to 305 °C under 1 sun irradiation. Furthermore, a soft

templating method is developed to synthesize CuZnAl nanosheets, which have excellent thermocatalytic MSR activity and stability due to their ultrathin thickness, large specific surface area, and uniform elemental distribution. Consequently, without consuming artificial energy, CuZnAl nanosheets combined with the $Bi_2Te_3$/Cu-based device exhibit a solar-heating MSR performance that is far beyond all of sunlight-driven methanol-based hydrogen production systems reported to date. Moreover, a scalable model is constructed in this work, and it successfully produces 23.27 m$^3$/day of hydrogen from MSR under 6 m$^2$ of outdoor sunlight irradiation in the spring.

## Results

### Using $Bi_2Te_3$/Cu to achieve a high solar-heating temperature.
$Bi_2Te_3$ is a typical photothermal material with a narrow band gap (<0.2 eV)[45,46] that can nearly fully absorb the solar spectrum (Supplementary Fig. 1a, b) and has a high carrier concentration of 0.84–1.11 × 10$^{19}$ cm$^{-3}$ (Supplementary Fig. 1c). Therefore, absorbed sunlight can be fully thermalized by this type of narrow-bandgap semiconductor via electron–phonon and electron–electron scattering[47]. For instance, Cheng et al. reported that $Bi_2Te_3$ could convert up to 99% of solar energy into heat energy[48]. To achieve a high solar irradiation temperature, besides the superior solar-to-thermal conversion, it is also necessary to localize the sunlight-converted heat energy in $Bi_2Te_3$ to reduce the amount of heat dissipation. Although vacuum protection was applied to cut off the heat conduction loss of the pure $Bi_2Te_3$ film, the 1-sun (1 kW m$^{-2}$) illumination temperature of the pure $Bi_2Te_3$ film was only 93 °C (Supplementary Fig. 1d). As a blackbody material (Fig. 1a)[49], the heat dissipation of the pure $Bi_2Te_3$ film includes not only the heat conduction loss but also, importantly, the violent heat radiation loss caused by infrared light (IR) radiation (IR emissivity of 0.91, as shown in the Supplementary Information)[50]. Therefore, minimizing the IR radiation of $Bi_2Te_3$ is the key to increasing its solar irradiation temperature. The IR light irradiated by $Bi_2Te_3$ is produced by lattice vibrations, and the lattice vibrations are proportional to the number of atoms in $Bi_2Te_3$[51]. From the physical principle, reducing the number of atoms in the $Bi_2Te_3$ structure can weaken IR radiation, so our strategy involves synthesizing a $Bi_2Te_3$ thin film to minimize the number of atoms and minimize the IR radiation, as shown in Fig. 1b. To achieve a low-IR radiation of the $Bi_2Te_3$ thin-film structure, the supports used to deposit the $Bi_2Te_3$ thin film also need to exhibit the low-IR radiation property. However, the supports used to deposit $Bi_2Te_3$ thin films are usually silicon wafer, which is also a typical blackbody material with strong IR radiation capability and cannot be used as a support for reducing the IR radiation of the whole $Bi_2Te_3$ thin-film structure[49]. Unlike blackbody materials, the highly conductive metal Cu contains a large number of nearly free electrons that can prevent the spillover of IR light[52], making Cu have near-zero IR radiation (~3% IR emissivity, Supplementary Fig. 2)[53,54]. Therefore, Cu film is selected as the support to synthesize $Bi_2Te_3$ thin films to make the hybrid have merits such as superior solar-to-thermal conversion from $Bi_2Te_3$ and low-IR radiation from Cu[55]. By controlling the deposition time, the thickness of the $Bi_2Te_3$ film on the Cu support was tuned to 3 μm (Fig. 1c), 100 nm (Fig. 1d), and 15 nm (Fig. 1e). The interface structure is shown in Supplementary Fig. 3. When the thickness of the $Bi_2Te_3$ film in $Bi_2Te_3$/Cu was 3 μm, the 1-sun irradiation temperature was 97 °C with vacuum protection (Fig. 1f), ~4 °C higher than that of pure $Bi_2Te_3$ (93 °C, Supplementary Fig. 1d) under the same conditions. Surprisingly, as the thickness of the $Bi_2Te_3$ film in $Bi_2Te_3$/Cu was reduced to 100 nm, the 1-sun irradiation temperature of vacuum-protected $Bi_2Te_3$/Cu increased sharply to 317 °C (Fig. 1g), which was not only 224 °C higher than that of pure $Bi_2Te_3$ but also 197 °C higher than the reported highest

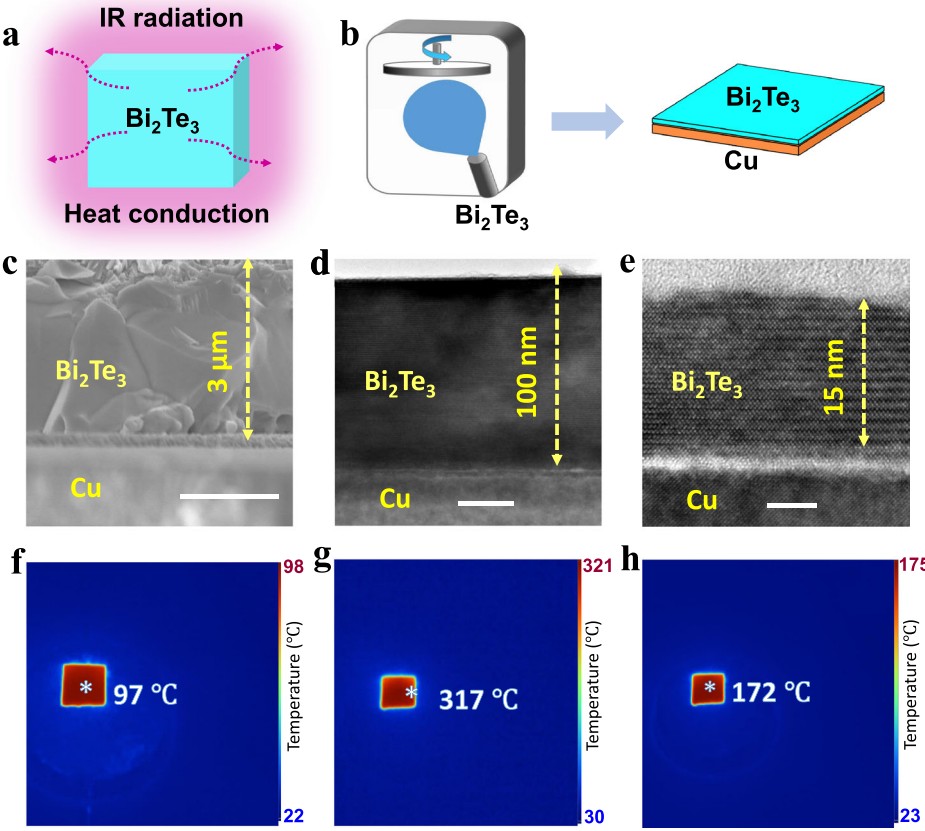

**Fig. 1 Heterostructure strategy used to increase the solar irradiation temperature of Bi₂Te₃. a** Map of the heat dissipation of the pure Bi₂Te₃ film. **b** Synthesis of the Bi₂Te₃ thin film on a Cu support (Bi₂Te₃/Cu). **c–e** SEM and TEM images of Bi₂Te₃/Cu with 3 µm-, 100 nm-, and 15 nm-thick Bi₂Te₃. **f–h** IR images of vacuum-protected Bi₂Te₃/Cu with 3 µm-, 100 nm-, and 15 nm-thick Bi₂Te₃. A CaF₂ glass fully covered each sample, and the vacuum degree of this equipment was $1.0 \times 10^{-3}$ Pa. The scale bars in **c**, **d**, and **e** are 1500, 30, and 5 nm, respectively.

1-sun irradiation temperature of photothermal materials (120 °C), as far as we know[25]. This indicates that this strategy is useful for improving the solar-heating temperature of photothermal materials. Furthermore, when we reduced the thickness of Bi₂Te₃ in Bi₂Te₃/Cu to 15 nm (Fig. 1e), the 1-sun irradiation temperature was only 172 °C (Fig. 1h), 145 °C lower than that of Bi₂Te₃/Cu with a Bi₂Te₃ thickness of 100 nm. Therefore, the thickness of Bi₂Te₃ has an important influence on the sunlight irradiation temperature of Bi₂Te₃/Cu.

**Thickness effect of Bi₂Te₃ and the device based on Bi₂Te₃/Cu.** To explain the effect of Bi₂Te₃ thin-film thickness on the sunlight irradiation temperature of Bi₂Te₃/Cu, we measured the light absorbed by the three Bi₂Te₃/Cu samples. For the 3 µm-, 100 nm-, and 15 nm-thick Bi₂Te₃ thin films in Bi₂Te₃/Cu, Fig. 2a–c show absorbances in the sunlight region (400–2000 nm) of ~94%, 89%, and 43%, respectively. Bi₂Te₃ has a narrow bandgap of <0.2 eV[45,46]; thus, sunlight has enough energy to excite electron transitions in Bi₂Te₃[56,57]. And, the film thickness of Bi₂Te₃ must be ≥100 nm to ensure more than 89% solar spectrum absorption. However, the absorption in the IR region was 4 and 5% when the thickness of the Bi₂Te₃ thin film in Bi₂Te₃/Cu was 15 and 100 nm, respectively (Fig. 2b, c), and it increased to 60% when the thickness of the Bi₂Te₃ thin film was further increased to 3 µm (Fig. 2a). As the absorptivity of light is equal to the emissivity of the corresponding light[54], the 60% IR absorption showed that the IR emissivity of Bi₂Te₃/Cu with a 3 µm-thick Bi₂Te₃ thin film is 60%, at least 10 times higher than that of Bi₂Te₃/Cu with 100 nm

(5%)- and 15 nm (4%)-thick Bi₂Te₃ thin films. For a more intuitive embodiment, we directly tested the IR radiation intensity (4–20 µm) of these samples heated to 93 °C. As shown in Fig. 2d, e, f, the IR radiation intensities in the range of 4 µm–20 µm are 248 W m⁻², 20.7 W m⁻², 16.6 W m⁻² for Bi₂Te₃/Cu with 3 µm-, 100 nm-, and 15 nm-thick Bi₂Te₃ thin films, respectively, significantly lower than the corresponding IR radiation of the pure Bi₂Te₃ film of 377 W m⁻² (Supplementary Fig. 1e). Summarizing the solar absorptivity and IR emissivity listed in Supplementary Table 1, the 100 nm-thick Bi₂Te₃ layer can synergistically absorb 89% of sunlight and emit 5% of IR radiation; in other words, this 100 nm-thick Bi₂Te₃ layer can absorb sunlight energy to the maximum extent and dissipate heat energy to the minimum extent so that the heat energy converted from sunlight is localized in the interior of the Bi₂Te₃ layer, resulting in a high sunlight irradiation temperature. This method is also suitable for other narrow-band gap semiconductors, such as Ti₂O₃[58], Cu₂Se[59], and Cu₂S[60]. When we synthesized ~200 nm-thick Ti₂O₃, Cu₂Se, and Cu₂S on a Cu support (Supplementary Fig. 4), the Ti₂O₃/Cu, Cu₂Se/Cu, Cu₂S/Cu heterostructures showed 1-sun irradiation temperatures of 295, 271, and 248 °C (Fig. 2g–i), respectively, obviously higher than the 1-sun heating temperatures of pure Ti₂O₃ (82 °C), Cu₂Se (79 °C), and Cu₂S (75 °C), as shown in Supplementary Fig. 5. Meanwhile, different thicknesses of Ti₂O₃, Cu₂Se, and Cu₂S on the Cu support were synthesized (Supplementary Fig. 6), and the corresponding IR images (Supplementary Fig. 7) showed that the temperatures of those samples were all higher than the 1-sun heating temperatures of pure Ti₂O₃ (82 °C), Cu₂Se (79 °C), and Cu₂S (75 °C), as shown in

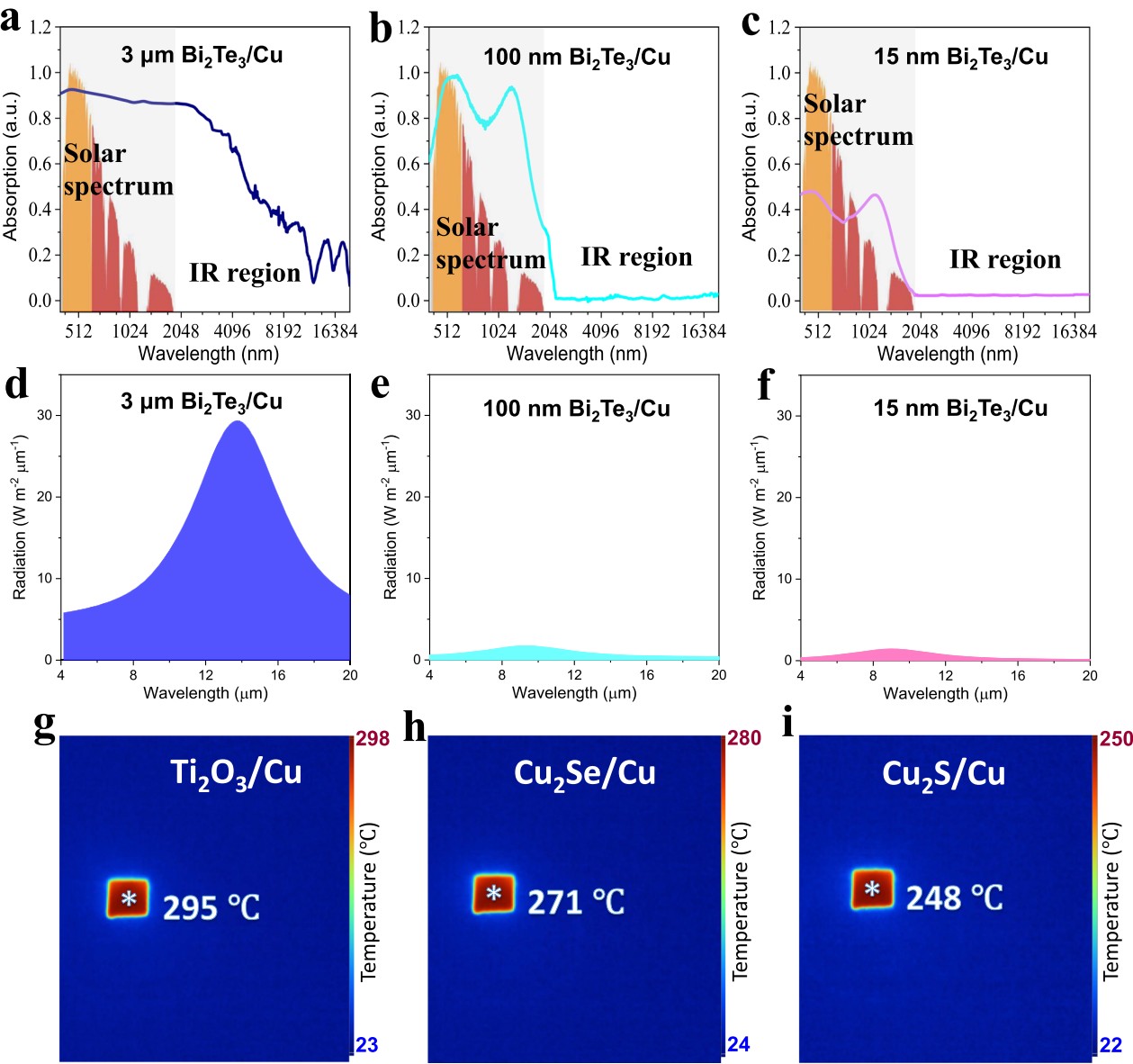

**Fig. 2 Thickness effect and universality of the heterostructure strategy. a–c** Normalized light absorption spectra of $Bi_2Te_3$/Cu with 3 μm-, 100 nm-, and 15 nm-thick $Bi_2Te_3$ layers. **d–f** IR radiation ranging from 4 μm to 20 μm for $Bi_2Te_3$/Cu with 3 μm-, 100 nm-, and 15 nm-thick $Bi_2Te_3$ layers at 93 °C. **g–i** IR images of vacuum-protected $Ti_2O_3$/Cu, $Cu_2Se$/Cu, $Cu_2S$/Cu heterostructures. A $CaF_2$ glass fully covered each sample, and the vacuum degree of this equipment was $1.0 \times 10^{-3}$ Pa.

Supplementary Fig. 5. These results confirm that the proper thickness is significant for narrow-bandgap semiconductors to have weak IR radiation while maintaining enough solar spectral absorption to achieve a high temperature in the device.

As shown in Fig. 3a, hybridization of the Cu layer, $Bi_2Te_3$ layer, vacuum layer, and glass layer was successively achieved on the outer surface of the reaction tube to form a device (named the $Bi_2Te_3$/Cu-based device). An optical image of a reaction tube is shown in Fig. 3b (Supplementary Fig. 8). Under 1-sun irradiation, the IR image shows that the inner temperature of the $Bi_2Te_3$/Cu-based device was as high as 307 °C (Fig. 3c). As we loaded the CuZnAl catalyst in the $Bi_2Te_3$/Cu-based device, Fig. 3d shows that the temperature of the CuZnAl catalyst was higher than 200 °C at 0.5-sun irradiation, and the temperature reached 305 °C under 1-sun irradiation. The temperature of the CuZnAl catalyst-loaded $Bi_2Te_3$/Cu-based device was 230 °C higher than that of the CuZnAl catalyst directly irradiated by 1 sun (Supplementary Fig. 9).

**Synthesis and characterization of the MSR catalyst: CuZnAl nanosheets.** With the device that can generate a high solar-heating temperature, we added commercial CuZnAl (C-CuZnAl, Supplementary Fig. 10) to the $Bi_2Te_3$/Cu-based device to test the sunlight-driven MSR performance. As shown in Supplementary Fig. 11, the 1 sun-driven MSR $H_2$ generation rate achieved with C-CuZnAl was 79.3 mmol $g^{-1}$ $h^{-1}$, which is much higher than the reported record photocatalytic MSR value (46.6 mmol $g^{-1}$ $h^{-1}$)[39], thus highlighting the importance of the $Bi_2Te_3$/Cu-based device. To achieve better sunlight-driven MSR performance, more efficient catalysts for MSR need to be developed. In this work, polyvinylpyrrolidone (PVP, K30) was selected as a surfactant to synthesize this type of CuZnAl catalyst[61]. PVP was mixed with the CuZnAl precursor as a homogeneous solution, and then, the $Na_2CO_3$ solution was dropped to precipitate CuZnAl oxides (Fig. 4a). In the precipitation process, PVP was able to guide anisotropic growth and prevent aggregation. Consequently, we successfully controlled the morphology of CuZnAl oxides by

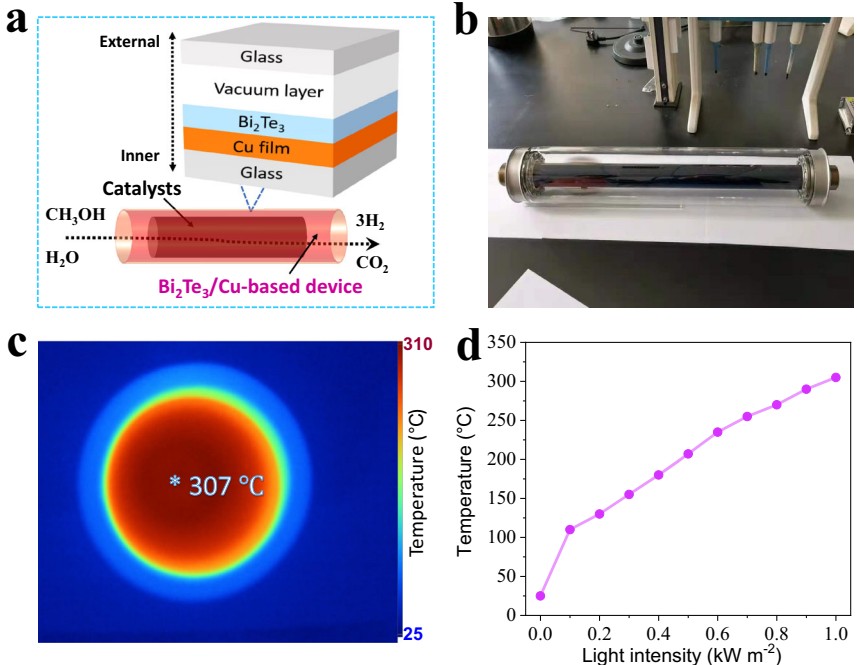

**Fig. 3 Bi$_2$Te$_3$/Cu-based device. a** Map of the Bi$_2$Te$_3$/Cu-based device loaded with catalysts for solar-heating MSR. **b** Photograph of the homemade Bi$_2$Te$_3$/Cu-based device. **c** Cross-sectional IR image of the Bi$_2$Te$_3$/Cu-based device under 1-sun irradiation. **d** Temperature of the CuZnAl catalyst loaded in the Bi$_2$Te$_3$/Cu-based device under different intensities of solar irradiation.

tuning the PVP amount (Supplementary Fig. 12, the optimized mass ratio of the PVP/CuZnAl precursor was 8). Supplementary Figure 13 shows that our synthesized sample could fully fill a 40 L bottle, revealing its scalable preparation. TEM imaging exhibits the porous nanosheet morphology of this sample (Fig. 4b), so we labeled the sample CuZnAl NS. The elemental mapping images in Fig. 4c reveal that Cu, Zn, and Al are homogeneously dispersed in the CuZnAl NS. Nanoparticles with diameters of <5 nm were observed in the HRTEM image, with distinguishable lattice fringes assigned to Cu and ZnO (Fig. 4d). The small sizes of Cu and ZnO provided more interfaces, so they were generally recognized as highly active sites for MSR[62]. The thickness of the CuZnAl NS was measured to be 3.2 nm (Fig. 4e). According to the nitrogen adsorption–desorption measurements (Fig. 4f), the CuZnAl NS exhibited a large specific surface area of 195.2 m$^2$ g$^{-1}$, four times larger than that of commercial C-CuZnAl (Supplementary Fig. 10), ensuring a large number of active sites for catalytic reactions. As a result, the hydrogen production rate of CuZnAl NS was 1.02 mol g$^{-1}$ h$^{-1}$ at 260 °C, quintupling the 0.2 mol g$^{-1}$ h$^{-1}$ hydrogen production rate of C-CuZnAl at 260 °C (Fig. 4g), and the methanol conversion rate was 5.28% (Supplementary Fig. 14). Additionally, we tested CuZnAl NS for 20 days, and the hydrogen production rate at 260 °C was maintained at ~1 mol g$^{-1}$ h$^{-1}$ (Supplementary Fig. 15), indicating the excellent stability of CuZnAl NS.

**Solar-heating MSR**. Ten grams of CuZnAl NS was placed in the Bi$_2$Te$_3$/Cu-based device, and 0.0471 m$^2$ of sunlight irradiation was the only energy source. When the sunlight density was higher than 0.5 sun, a clear hydrogen signal appeared for the CuZnAl NS loaded in the Bi$_2$Te$_3$/Cu-based device, and the hydrogen generation rate increased to 3.1 mol h$^{-1}$ (Fig. 5a) under 1 sun irradiation, corresponding to 310 mmol g$^{-1}$ h$^{-1}$ and a methanol conversion rate of 45.34% (Supplementary Fig. 16). Meanwhile, the temperature of the CuZnAl NS loaded in the Bi$_2$Te$_3$/Cu-based device during solar-heating MSR is shown in Supplementary

Fig. 17. Comparatively, MSR showed a hydrogen production rate of zero, as the CuZnAl NS were directly irradiated by 1-sun irradiation without the device (Fig. 5a). Since sunlight is the only energy source of the MSR catalytic reaction in solar-heating catalysis, as in the photocatalysis reaction, we listed the state-of-the-art sunlight-driven hydrogen production rates in Fig. 5b and Table 1 for comparison with our data. Figure 5b and Table 1 confirm that our tested activity was at least 6 times the activity of the best sunlight-driven hydrogen production systems in the reported literature, e.g., Ni/CdS (46.6 mmol g$^{-1}$ h$^{-1}$)[39], NiS$_x$/Cd$_{0.5}$Zn$_{0.5}$S (44.6 mmol g$^{-1}$ h$^{-1}$)[34], Mg-black TiO$_2$ (43.1 mmol g$^{-1}$ h$^{-1}$)[37], Ni(II)/CdS (43 mmol g$^{-1}$ h$^{-1}$)[63], NiO/LaNaTaO$_3$ (38.4 mmol g$^{-1}$ h$^{-1}$)[64], BP/Bi$_2$WO$_6$ (21.0 mmol g$^{-1}$ h$^{-1}$)[65], C$_3$N$_4$ (19 mmol g$^{-1}$ h$^{-1}$)[66], CdS/2H-MoS$_2$ (16.6 mmol g$^{-1}$ h$^{-1}$)[67], N-doped black TiO$_2$ (15 mmol g$^{-1}$ h$^{-1}$)[68], Sr2MgSi2O7:Eu$^{2+}$ (14 mmol g$^{-1}$ h$^{-1}$)[69], and black TiO$_2$ (10 mmol g$^{-1}$ h$^{-1}$)[36]. Based on the experimental data, the solar-to-hydrogen (STH) conversion efficiencies of CuZnAl NS in the Bi$_2$Te$_3$/Cu-based device were calculated to be 31.9% and 30.1% under 0.9- and 1-sun irradiation, respectively (Fig. 5c). Note that the STH of our solar heating MSR is beyond the theoretical STH limit of photocatalytic MSR achieved through the route of photon-photogenerated electrons and holes-chemicals[70]. Ishii et al. reported that the average energy of photons in the solar spectrum is ~1 eV[71]. However, the reaction enthalpy of MSR is 0.086 eV$_{per H}$ (1/3 CH$_3$OH (l) + 1/3 H$_2$O (l) → H$_2$ (g) + 1/3 CO$_2$ (g); detailed calculation shown in the Methods). Therefore, the STH ceiling of photocatalytic MSR under 1-sun illumination is 8.6% (0.086 eV/ 1 eV), equivalent to ~1/3 of the STH of our solar heating MSR system (30.1%) under 1-sun irradiation. This work reveals that solar heating catalysis via a solar-thermal energy-chemical route has an incomparable advantage over photocatalysis in reactions with a low energy barrier. Therefore, the Bi$_2$Te$_3$/Cu-based device with CuZnAl NS opens a pathway for efficiently achieving solar-driven hydrogen generation from methanol and water, in which solar heating is the only energy source used and has no energy

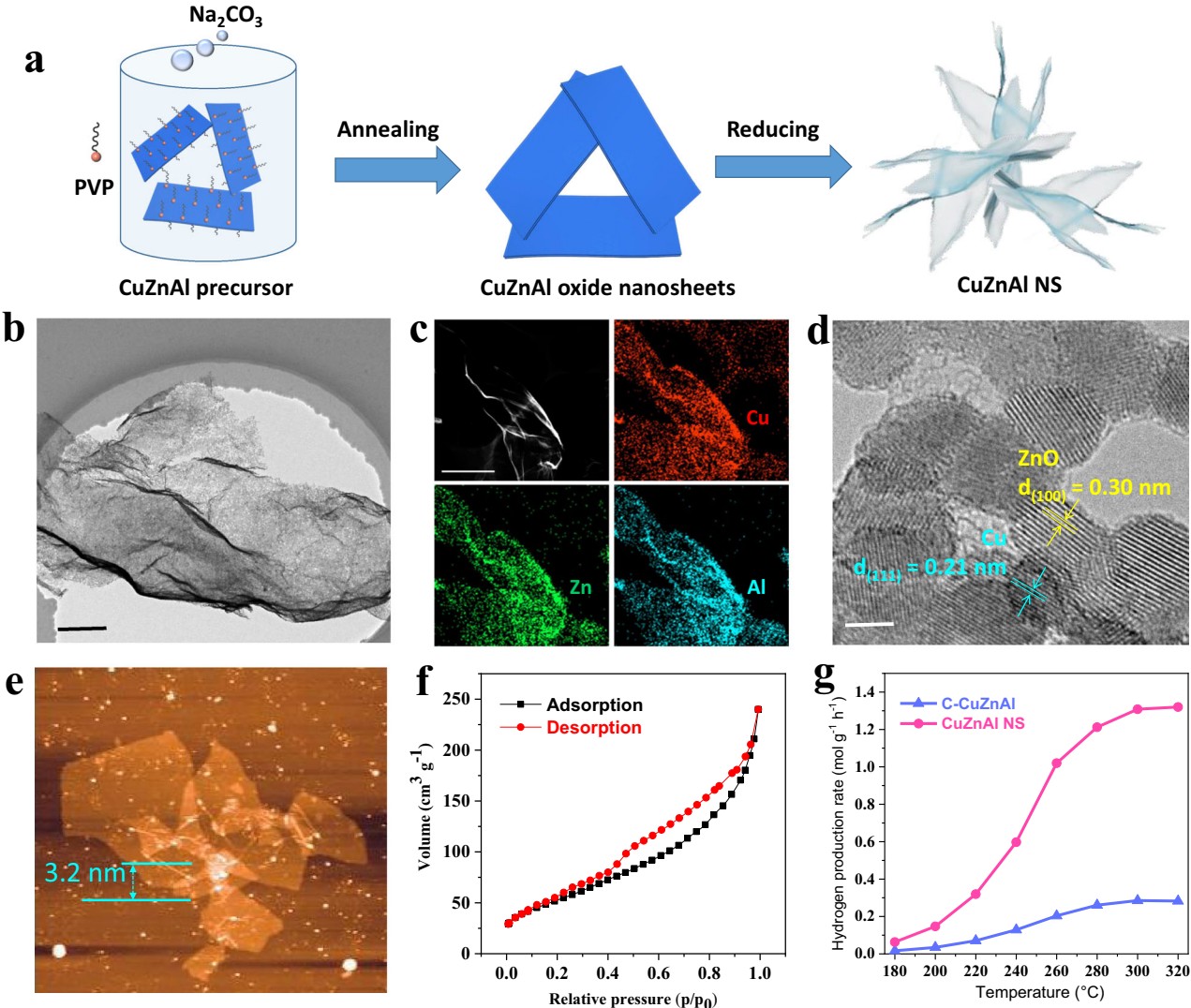

**Fig. 4 Preparation and characterization of CuZnAl NS. a** Schematic of the synthesis process for CuZnAl NS. TEM image **b**, elemental mapping image **c**, HRTEM image **d**, AFM image **e**, and nitrogen adsorption and desorption isotherm **f** of CuZnAl NS. **g** Hydrogen production rate from the MSR of CuZnAl NS and C-CuZnAl at different temperatures. Test conditions: 10 mg of catalyst, 50 sccm of Ar, and 0.1 sccm of methanol (the volume ratio of methanol to water = 1:1.3). The scale bars in **b**, **c**, and **d** are 200, 500, and 2 nm, respectively. CuZnAl NS and C-CuZnAl are abbreviations for CuZnAl nanosheets and commercial CuZnAl, respectively.

supply from artificial input is needed, meaning that artificial energy is not consumed. In addition to the high hydrogen production rate, the ratio of $CO_2$ to $CO_2 + CO$ in our solar-heating MSR strategy was higher than 99.2% under sunlight irradiation (Supplementary Fig. 18), indicating a low CO concentration in the hydrogen-producing process.

To test the capability of the $Bi_2Te_3/Cu$-based device for large-scale production (Supplementary Fig. 19), we prepared a scalable system, as shown in Fig. 5d. Its outdoor test was performed on 8 April 2021, with an ambient temperature of 6–21 °C and a sunlight intensity of 0.15–0.52 kW m$^{-2}$ (Fig. 5e) in the daytime in Baoding City of Hebei Province, China. To make the system work well in the morning and evening, the system was equipped with a parabolic reflector with a 6 m$^2$ irradiation area to concentrate sunlight to moderate its solar-heating MSR ability (Supplementary Movie 1). As shown in Fig. 5f, MSR took place at 8:00 A.M. with a hydrogen production rate of 1.61 m$^3$ h$^{-1}$. After that, the rate rose sharply, and the hydrogen production rate reached a peak of 3.56 m$^3$ h$^{-1}$ at 12:00 P.M. and then gradually decreased to 1.19 m$^3$ h$^{-1}$ at 17:00 P.M. The total amount of hydrogen produced daily was up to 23.27 m$^3$ under ambient sunlight irradiation, showing the potential of industrialization.

## Discussion

In this work, we propose a solar-heating catalysis mode as a distinctive type of artificial-energy-input-free catalytic system. A heterostructure consisting of black photothermal materials, used to fully absorb sunlight, and a Cu support, used to weaken IR radiation, was used to increase the solar-heating temperature of photothermal materials. Taking $Bi_2Te_3$ as an example, we found that the solar-spectrum absorption and IR radiation of the $Bi_2Te_3$ film depended on the thickness of the $Bi_2Te_3$ film. Consequently, hybridization of the 100 nm-thick $Bi_2Te_3$ film with a Cu support showed a 1 sun-heating temperature of 317 °C with vacuum protection, 224 °C higher than that of pure $Bi_2Te_3$ under the same conditions. This strategy is widely used in narrow-band gap materials, and the hybrids of $Ti_2O_3/Cu$, $Cu_2Se/Cu$, and $Cu_2S/Cu$ exhibited 1 sun-heating temperatures of 295, 271, and 248 °C, respectively. A PVP-capped coprecipitation method was modified

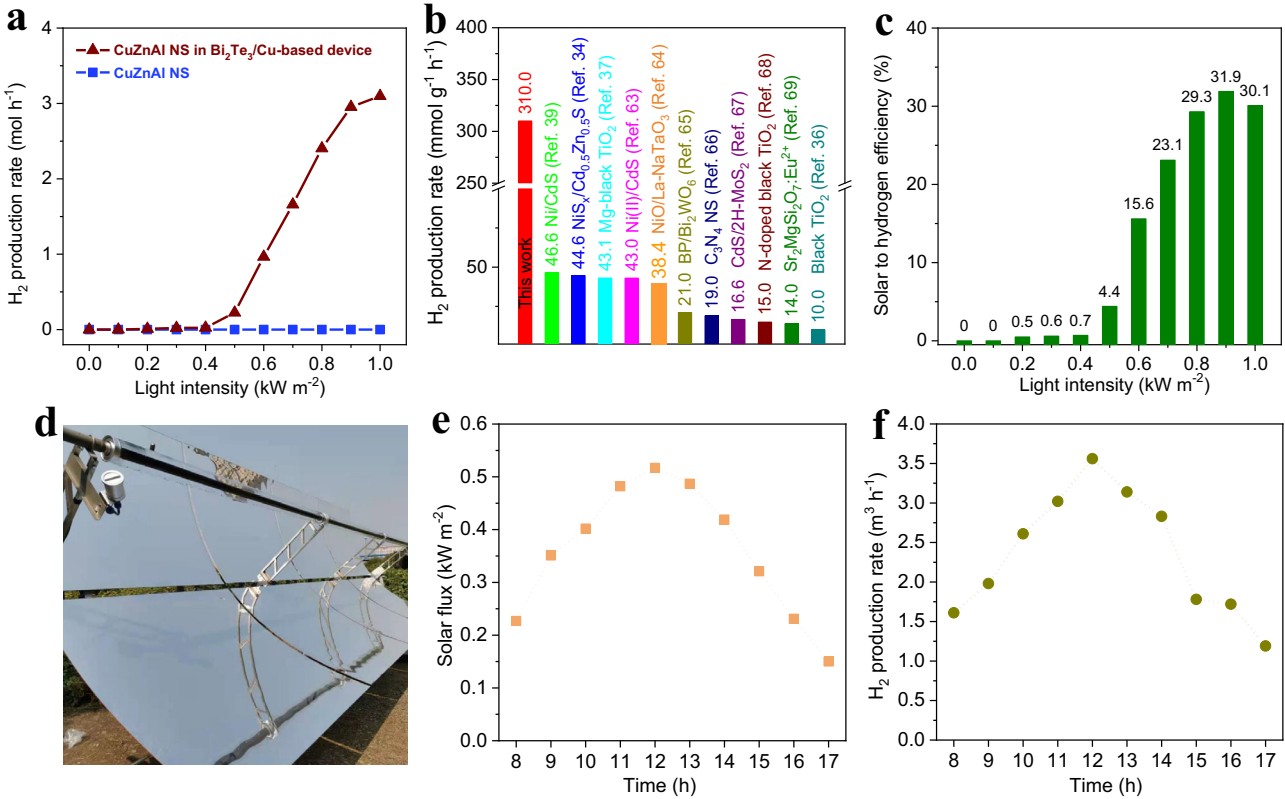

**Fig. 5 Solar-heating MSR performance of CuZnAl NS with a Bi₂Te₃/Cu-based device. a** Hydrogen production rates from the MSR of CuZnAl NS loaded in the Bi₂Te₃/Cu-based device and the device without CuZnAl NS under different sunlight irradiation. **b** Visual contrast diagram of the hydrogen production rates achieved under 1 sun illumination by the Bi₂Te₃/Cu-based device loaded with CuZnAl NS and those of other photocatalysts. **c** STH efficiency of MSR achieved by the Bi₂Te₃/Cu-based device loaded with CuZnAl NS under different intensities of solar irradiation. **d** Optical image of the solar-heating system used for hydrogen production from MSR under ambient sunlight irradiation. **e** Solar flux on April 08, 2021, in Baoding City, China. **f** MSR hydrogen production rate as a function of time under ambient sunlight irradiation. CuZnAl NS is the abbreviation for CuZnAl nanosheets.

**Table 1 Sunlight-driven hydrogen generation from the MSR of CuZnAl NS loaded in a Bi₂Te₃/Cu-based device in comparison with the reported advanced sunlight-driven hydrogen generation systems.**

| Catalysts | Light source | Reagents | T (°C) | H₂ rate mmol g⁻¹ h⁻¹ | Refs. |
|---|---|---|---|---|---|
| CuZnAl NS | Xe lamp | $CH_3OH$, $H_2O$ | 25 | 310 | This work |
| Ni/CdS | Xe lamp | 2-propanol, $H_2O$ | 20 | 46.6 | 39 |
| NiSₓ/Cd₀.₅Zn₀.₅S | Xe lamp | $Na_2S$ $Na_2SO_3$, $H_2O$ | 25 | 44.6 | 34 |
| Mg-black $TiO_2$ | Xe lamp | $CH_3OH$, $H_2O$ | 25 | 43.1 | 37 |
| Ni(II)/CdS | Xe lamp | $Na_2S$ $Na_2SO_3$, $H_2O$ | 25 | 43 | 63 |
| $NiO/LaNaTaO_3$ | Hg lamp | $H_2O$ | 25 | 38.4 | 64 |
| $BP/Bi_2WO_6$ | Xe lamp | NO $CH_3OH$, $H_2O$ | 25 | 21.0 | 65 |
| $C_3N_4$ NS | Xe lamp | $K_2HPO_4$, TEOA $H_2O$ | 25 | 19 | 66 |
| $CdS/2H-MoS_2$ | Xe lamp | $Na_2S$ $Na_2SO_3$, $H_2O$ | 25 | 16.6 | 67 |
| N-doped black $TiO_2$ | Xe lamp | $CH_3OH$, $H_2O$ | 25 | 15.0 | 68 |
| $Sr_2MgSi_2O_7:Eu^{2+}$ | Hg lamp | $CH_3OH$, $H_2O$ | 25 | 14 | 69 |
| Black $TiO_2$ | Xe lamp | $CH_3OH$, $H_2O$ | 25 | 10 | 36 |

*T* test temperature.

and used to synthesize CuZnAl nanosheets (CuZnAl NS) on a large scale with a thickness of 3.2 nm, a larger specific surface area of 195.2 m² g⁻¹, and a 5-nm Cu nanoparticle as benchmark catalysts for MSR. Finally, based on Bi₂Te₃/Cu, we synthesized a reaction device in which CuZnAl NS were heated to 305 °C under 1-sun irradiation, and 0.15 m² of 1 sun-heated MSR showed a hydrogen production rate of 3.1 mol h⁻¹, at least 6 times higher than that reported for sunlight-driven hydrogen production systems, with an STH efficiency of 30.1% and 20-day stability.

Moreover, an industrial demo of our system driven by 6 m² of outdoor sunlight was able to generate 23.27 m³/day of H₂ from MSR. In these systems, the energy source used is only solar heating, and no other artificial energy is consumed.

## Methods

**Deposition of the Bi₂Te₃ film and synthesis of devices.** SP-0707AS magnetron sputtering was used to deposit the Bi₂Te₃ film at a vacuum pressure lower than 7.0 × 10⁻³ Pa, and a 4-axis rotation system was used to rotate the bases. Bi₂Te₃ and

Cu were used as targets; the working gas was Ar with 99.99% purity. The bases used in Figs. 1–3 and 5 were Cu films 20 × 20 × 0.1 mm in size, Cu films 20 × 20 × 0.1 mm in size, reaction tubes 250 mm in length and 30 mm in diameter, and stainless steel tubes 2000 mm in length and 42 mm in diameter, respectively. Before the deposition process, the bases were subsequently washed with deionized water, acetone, and ethanol.

(1) For the deposition of the $Bi_2Te_3$ film on the Cu film, glow discharge was applied to clean the Cu film, and then, the $Bi_2Te_3$ film was deposited. Finally, the sample was removed after passive cooling. The power was 5 kW, the sputtering pressure was $9 \times 10^{-2}$ Pa, the bias voltage was 150 V, the sputtering temperature was 70 °C, and the sputtering times were 1 min and 6 min for $Bi_2Te_3$ films with 15 nm and 100 nm thicknesses, respectively. For the 3 μm-thick $Bi_2Te_3$ film, the sputtering pressure was $7 \times 10^{-1}$ Pa, and the sputtering time was 15 min.

(2) For the synthesis of the $Bi_2Te_3$/Cu-based device involving the deposition of the Cu substrate and $Bi_2Te_3$ film on the reaction tube, glow discharge was first applied to clean the glass tube. Then, the Cu layer was deposited by the Cu target and the $Bi_2Te_3$ film was deposited by the $Bi_2Te_3$ target in an orderly manner, with the sample being removed after passive cooling. The power was 5 kW, the sputtering pressure was $9 \times 10^{-2}$ Pa, the bias voltage was 150 V, the sputtering temperature was 70 °C, and the sputtering times for the Cu layer and $Bi_2Te_3$ film were 12 min and 6 min, respectively. The thickness of the Cu layer was ~10 μm. The followed antireflection film and glass vacuum layer were provided by Hebei Scientist Research Experimental and Equipment Trade Co., Ltd. with a $1 \times 10^{-3}$ Pa pressure. The final product is shown in Fig. 3b.

(3) For the synthesis of the device shown in Fig. 5d, involving the deposition of the Cu substrate and $Bi_2Te_3$ film on a stainless steel tube, glow discharge was first performed to clean stainless steel tube, and then, the Cu layer was deposited by the Cu target and the $Bi_2Te_3$ film was deposited by the $Bi_2Te_3$ target in an orderly manner. Finally, the sample was removed after passive cooling. The power was 5 kW, the sputtering pressure was $9 \times 10^{-2}$ Pa, the bias voltage was 150 V, the sputtering temperature was 70 °C, and the sputtering times for the Cu layer and $Bi_2Te_3$ film were 12 min and 6 min, respectively. The followed antireflection film and glass vacuum layer were provided by Hebei Scientist Research Experimental and Equipment Trade Co., Ltd. at a $1 \times 10^{-3}$ Pa pressure. The tubes were welded together to form a reaction tube 6 m in length for the outdoor test.

**Chemicals for catalysts**. Commercial copper nitrate ($Cu(NO_3)_2$), zinc nitrate hydrate ($Zn(NO_3)_2 \cdot 6H_2O$), aluminum nitrate hydrate ($Al(NO_3)_3 \cdot 9H_2O$), poly-vinylpyrrolidone (PVP, K30), sodium borate, and sodium carbonate were purchased from Sinopharm Co., Ltd. The chemicals were all used without any further treatment.

**Catalyst preparation (CuZnAl NS, C-CuZnAl)**. A total of 375.7 g of $Cu(NO_3)_2$, 297.4 g of $Zn(NO_3)_2 \cdot 6H_2O$, and 125.1 g of $Al(NO_3)_3 \cdot 9H_2O$ were dissolved in 20 L of deionized water (containing 760 g of sodium borate). Then, 32 L of 150 mg mL$^{-1}$ PVP aqueous solution was added to the above solution. The mixture solution was denoted solution A. Then, 0.2 M aqueous $Na_2CO_3$ (10 L) was prepared and denoted solution B. Solution B was slowly added to solution A under stirring at 65 °C. The mixture solution was further stirred for 1 h and aged for 16 h at 65 °C. The precipitate was collected by centrifugation, washed with water three times, and dried by freeze-drying. Then, CuZnAl NS were obtained by calcination at 400 °C in air for 6 h and reduced in 10% $H_2$/Ar at 300 °C for 10 h.

C-CuZnAl was prepared by the coprecipitation method. Typically, 37.512 g of $Cu(NO_3)_2$, 29.749 g of $Zn(NO_3)_2$ $6H_2O$, and 12.504 g of $Al(NO_3)_3$ $9H_2O$ were dissolved in 200 mL of deionized water. After stirring for 1 h, the resulting solution and $Na_2CO_3$ aqueous solution (0.2 M; 1 L) were added dropwise and stirred at 65 °C for 1 h. After holding at 65 °C for 4 h and ageing for 7 h, the resulting precipitate was washed several times with deionized water and then fast-frozen in liquid nitrogen. The frozen cube was freeze-dried at −55 °C and then calcinated in air at 400 °C for 6 h to obtain Cu-Zn-Al-based oxides. Finally, the products were reduced to 10% $H_2$/Ar at 300 °C for 10 h and named C-CuZnAl.

We tested the density of both CuZnAl NS and C-CuZnAl. The density of the CuZnAl NS powder was ~0.12 g cm$^{-3}$, and that of the C-CuZnAl powder was ~0.795 g cm$^{-3}$.

**STH calculation**. For the STH calculation, a $Bi_2Te_3$/Cu-based device with an irradiation area of 0.0471 m$^2$ was used in this experiment, and 10 g of CuZnAl NS fully filled the inner space of this device. In this test, methanol and water were mixed as a solution with a methanol to water volume ratio = 1:1.6, and the mixed solution was then pumped into the system. To analyse the hydrogen gas product, we first removed $CO_2$ from the produced gas through a NaOH solution (5 M), and a flowmeter (MV-192-H2, Bronkhorst) was used to measure the flow rate, which was recognized as the rate of hydrogen production.

The STH efficiency of hydrogen generation from MSR was calculated as follows:

$$STH = (\Delta H \times \varepsilon / 24.5)/(I \times S \times 3600) \qquad (1)$$

The enthalpy change energies of methanol, $CO_2$, $H_2$, and $H_2O$ were −201.083, −393.505, 0, and −241.818 kJ mol$^{-1}$, respectively.

ΔH is the reaction enthalpy change of methanol dehydrogenation (1/3 $CH_3OH$ (l) + 1/3 $H_2O$ (l)→ $H_2$ (g) + 1/3 $CO_2$ (g), ΔH = 16.47 KJ mol$^{-1}$), ε (L) is the amount of $H_2$ generated per hour detected by a flowmeter (MV-192-H2), I is the light intensity (kW m$^{-2}$), and S is the irradiated area of catalysts (0.0471 m$^{-2}$). The calculation details and data are shown in Supplementary Fig. 20.

As 1 eV = $1.6 \times 10^{-19}$ J, the ΔH per $H_2$ was calculated to be 16.47 KJ/ $(1.6 \times 10^{-19} * 6.02 \times 10^{23})$ = 0.171 eV; therefore, the ΔH per H was calculated to be 0.171 eV/2 = 0.086 eV.

## Data availability
The data that support the findings of this study are available from the corresponding authors upon reasonable request.

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

## Acknowledgements

This work was supported by the Hebei Natural Science Foundation (Grant No. B2021201074), the Hebei Provincial Department of Science and Technology (Grant No. 216Z4303G), Hebei Education Department (Grant No. BJ2019016), the Advanced Talents Incubation Program of Hebei University (Grant Nos. 521000981248 and 8012605), the National Nature Science Foundation of China (Grant Nos. 51702078, 61774053, 61504036, 51972094, and 51971157), the Natural Science Foundation of Hebei Province (Grant Nos. B2021201034, F2019201446, and F2018201058), the National Key Research and Development Program of China (2018YFB1500503-02), the Scientific Research Foundation of Hebei Agricultural University (YJ201939), and the Tianjin Science Fund for Distinguished Young Scholars (19JCJQJC61800). Thank you for the TEM technical support provided by the Microanalysis Center, College of Physics Science and Technology, Hebei University.

## Author contributions

Y.L., S.W. and J.L. conceived the project and contributed to the design of the experiments and analysis of the data. Y.L. and D.Y. prepared and characterized the Bi2Te3-based device. X.B., Bo.L. and F.Z. prepared and characterized the catalyst. Ba.L. and G.F. provided optical advice. X.S. conducted SEM and TEM. Y.L. and J.L. wrote the paper. All the authors discussed the results and commented on the manuscript.

## Competing interests

The authors declare no competing interests.
