## [Peer Review File · Nature Communications]

Title: General Heterostructure Strategy of Photothermal Materials for Scalable Solar-heating Hydrogen Production Without the Consumption of Artificial EnergyREVIEWER COMMENTS

Reviewer #1 (Remarks to the Author):

Yaguang et al. conducted a study on the “Photothermal Materials for Scalable Solar-heating Hydrogen Production Without Consuming Artificial Energy”. In this study the demo reactor and the high H₂ generation rate is worth of affirmation. However, the major innovation of this study focuses on the device and process demonstration. Therefore, I evaluate this work not suitable for Nat. Comm, and I suggest submitting it on an Engineering related journal.

First, the authors should clarify the methanol water reforming in this study is a photocatalytic reaction, thermal catalytic reaction or combined reaction. Because most of photocatalysis reactions are “Without Consuming Artificial Energy”. The authors “only” compared this study with other sunlight-driven hydrogen production in Fig. 5b and Tab. 1. It is worth noting that, as the Solar Heating on the Photothermal Materials created a high temperature environment, which high enough (above 300 oC) to trigger a thermal catalysis reaction. Especially the CuZnAl is a high-performance methanol water reforming catalyst. Therefore, I suggest the authors clarifying their catalysis system and setting up a reasonable comparison with another studies. Furthermore, to make this reaction running, a “Artificial Energy” driven pump is needed.

Second, CuZnAl catalyst synthesis does not have significant novelty (<https://doi.org/10.1016/j.apsusc.2019.144364>). And the author should show the bulk density of both CuZnAl NS and C-CuZnAl, because it affects the catalyst packing volume in the reactor.

Third, in this study, the H₂ production rates were depending on the Photothermal Materials, hybrid of Bi₂Te₃ and Cu showing the best performance. As show in the references listed in this manuscript, the Bi₂Te₃ as photothermal materials have been heavily study. And novelty of hybrid of Bi₂Te₃ and Cu preparation can not reach the level of Nat. Comm.

Here some comments on the details:

1) In the introduction section: “But, hydrogen generation from methanol and water by the methanol steam reforming ($\text{CH}_3\text{OH} + \text{H}_2\text{O} = \text{CO}_2 + 3\text{H}_2$, MSR) requires large-scale external energy input (3.2 kJ energy for 1 mol of H₂).33,35,36” I did not find this energy number (3.2 kJ energy) in these studies.

Furthermore, when we talk about the external energy input, we use the enthalpy. And ΔH of methanol water reforming is +49.7 kJ/mol. (Renewable and Sustainable Energy Reviews 29(2014)355–368), based on this the authors should provide detailed calculation of STH calculation.

2) Table 1, the authors should list reaction conditions such as reaction temperature.

3) The authors should list the methanol conversion.

4) Methanol water reforming is endothermic reaction, the authors should measure the temperature during the reaction.

5) In Figure 1, The thickness of Bi₂Te₃ affects the temperature. I suggest try to measure the temperature of Ti₂O₃/Cu, Cu₂Se/Cu, Cu₂S/Cu with different thickness, then compare the highest results.

6) The Figure 2 d is confusing.

7) Figure 4 b is not meaningful. I suggest the authors label the scale bars in Figure c d e.

Reviewer #2 (Remarks to the Author):

The work by Li, Wang, Luo, and coworkers demonstrate a novel photothermal material for converting sunlight into non-artificial heat with unprecedented high temperatures reached. The authors demonstrate the use of this material by performing methanol steam reforming (MSR) at 260 degrees Celcius entirely driven by solar heating, which is very impressive and highly interesting in my mind. However, solar photothermal driven thermochemical MSR reactions is not new, as the authors point out themselves. Nevertheless, the presented work does provide an improvement of approximately an order of magnitude compared to the current state-of-the-art, and their device is stable for more than 20 days. I find it confusing, though, that the authors apparently go on to synthesize a new MSR catalyst to confirm the use of their novel photothermal material, instead of demonstrating it with already published thermally driven catalytic reactions. How does their approach validate the superiority of Bi₂Te₃/Cu over other solar-to-thermal conversion devices? And how do we really compare the results when both the catalyst and the photothermal device are new? If I am to assess the overall activity and stability of the MSR, the work is, in my view, of high significance, but does not warrant publication in Nature Communications in its current form. However, the work should be re-assessed after the following considerations:

- 1) The concept of "Methanol economy" by Olah ought to be mentioned.
- 2) If the photothermal device is merely developed to provide breakthrough-level performing solar driven MSR, I suggest to further optimize the results to obtain a more significant improvement over state-of-the-art. If the MSR is included to merely highlighting the power of the photothermal device, I suggest to test and compare outcome when using known catalysts for MSR.
- 3) The authors employ MSR as a benchmarking reaction for conducting a thermally driven reaction at a high temperature above 120 degrees Celcius using their novel photothermal device. In my mind, it would be pertinent at least to acknowledge that there indeed exist low-temperature methanol reforming protocols performing well below 120 degrees Celcius (e.g. Beller Nature 2013, 495, 85 as well as Trincado and Grützmacher Nature Chemistry 2013, 5, 342), and to explain why it is pertinent to develop a sun-to-heat device for high-temperature MSR when low-temperature systems have already been developed on for almost a decade. An explanation becomes even more relevant when considering that PEM fuel cells generates an excess of heat that the low-temperature MSR potentially could exploit.
- 4) The authors mention that their system produces a low concentration of CO; however, 0.8% of CO (of CO₂ and CO) is not a low concentration for PEM fuel cells. Please consider this more carefully, especially in the light of that they mention how this system may be used for powering a vehicle.
- 5) It would be instrumental with some more pictures that clearly show the function and setup of the device, eg of the outdoor setup.

Reviewer #3 (Remarks to the Author):

In this manuscript, the authors propose to combine black photothermal materials and infrared insulating

materials, so that an elevated solar-heating temperature can be achieved. As an example, the Bi₂Te₃/Cu hybrid showed a 1 Sun-heating temperature of 317 °C. In addition, a reaction device based on Bi₂Te₃/Cu was assembled, which was able to heat a catalyst of CuOx/ZnO/Al₂O₃ nanosheets to 305 °C and showed a 1 Sun driven hydrogen production rate of 75.9 L/h from methanol and water. These results are impressive, and the following issues need addressed before possible publication of this manuscript.

1 When the thickness of Bi₂Te₃ layer was 100 nm, the Bi₂Te₃/Cu exhibited the highest 1 Sun irradiated temperature of 317 °C, higher than those for 15nm and 3 micron thick Bi₂Te₃. Is this an optimized result? What is the reason behind?

2 What is the fine structure of the Bi₂Te₃ layer? As seen from the TEM images, there seems to be difference between the structures of the Bi₂Te₃ layer with different thicknesses. Is the heating performance of the hybrid only thickness-related, or influenced by the structure of Bi₂Te₃?

3 What is the Bi₂Te₃/Cu interface structure? How does it affect the Sun-heating performance?

4 An illustration and corresponding discussion clearly showing the working mechanism of this hybrid should be presented, i.e. how is the sun light irradiation transferred to thermal energy efficiently through the Bi₂Te₃ layer, Cu and their interface? What is the key physics of this hybrid design?

REVIEWER COMMENTS

Reviewer #1 (Remarks to the Author):

Yaguang et al. conducted a study on the “Photothermal Materials for Scalable Solar-heating Hydrogen Production Without Consuming Artificial Energy”. In this study the demo reactor and the high H₂ generation rate is worth of affirmation. However, the major innovation of this study focuses on the device and process demonstration. Therefore, I evaluate this work not suitable for Nat. Comm, and I suggest submitting it on an Engineering related journal.

First, the authors should clarify the methanol water reforming in this study is a photocatalytic reaction, thermal catalytic reaction or combined reaction.

Response: The authors thank for the reviewer’s constructive comment. The methanol water reforming is thermal catalytic reaction in solar heating catalysis, as the catalysts loaded in the device can not absorb the sunlight, which is depicted in Supplementary Fig. 8.

Supplementary Fig. 8 The picture of catalysts loaded in Bi₂Te₃/Cu based device.

Because most of photocatalysis reactions are “Without Consuming Artificial Energy”. The authors “only” compared this study with other sunlight-driven hydrogen production in Fig. 5b and Tab. 1. It is worth noting that, as the Solar Heating on the Photothermal Materials created a high temperature environment, which high enough (above 300 °C) to trigger a thermal catalysis reaction. Especially the CuZnAl is a high-performance methanol water reforming catalyst. Therefore, I suggest the authors clarifying their catalysis system and setting up a reasonable comparison with another studies.

Response: The authors thank for the reviewer’s comment.

The heat energy used for solar heating MSR is achieved only from solar by the photothermal conversion of Bi₂Te₃/Cu based device. In comparison, the energy

consumed by traditional thermocatalytic MSR is artificially input electricity or fossil energy, while the energy consumed by photocatalytic MSR is sunlight. Therefore, the comparison between solar heating MSR with photocatalytic MSR is fair. The purpose of this comparison is to show that the path of solar-heat energy-chemicals in our work is more efficient than the path of solar-photogenerated carriers-chemicals in photocatalytic MSR. We can use the path of solar-heat energy-chemicals to drive the low energy barrier reactions (such as MSR) more efficiently.

Furthermore, to make this reaction running, a “Artificial Energy” driven pump is needed.

Response: The authors thank for the reviewer’s comment.

Firstly, what we call the demand for artificial energy mainly refers to the catalysis (MSR) itself, which does not involve peripheral pumping and other auxiliary processes. We have changed the expression in the revised manuscript.

“Promoting industrial catalysis without artificial energy input is of great significance for human beings.

Therefore, constructing artificial-energy-input-free catalysis is the key to human sustainable development.”

Secondly, the efficiency of our solar heating MSR is much higher than that of photocatalysis, and the consumption rate of methanol aqueous solution is too fast, which requires continuous pumping to provide methanol aqueous solution. If photocatalysis consumes all methanol, photocatalysis also requires a pumping process to add methanol.

Second, CuZnAl catalyst synthesis does not have significant novelty (<https://doi.org/10.1016/j.apsusc.2019.144364>). And the author should show the bulk density of both CuZnAl NS and C-CuZnAl, because it affects the catalyst packing volume in the reactor.

Response: The authors thank for the reviewer’s comment. The CuZnAl catalyst

synthesis method shown in <https://doi.org/10.1016/j.apsusc.2019.144364> used AAO/Al as the substrate to grow CuZnAl nanosheets. As we all know, using AAO/Al to synthesize catalyst cannot produce catalyst on large scale, that is, the kg scale production of catalyst cannot be carried out. However, our method can mass-produce CuZnAl nanosheets in kg level directly, which is the biggest highlight of our method for catalysts synthesis.

We tested the density of both CuZnAl NS and C-CuZnAl, and the density of CuZnAl NS powder is about 0.12g cm^{-3} and the one of C-CuZnAl powder is about 0.795g cm^{-3} . We have added the data in the revised manuscript.

“We tested the density of both CuZnAl NS and C-CuZnAl, the density of CuZnAl NS powder was $\sim 0.12\text{g cm}^{-3}$ and C-CuZnAl powder was $\sim 0.795\text{g cm}^{-3}$.”

Third, in this study, the H_2 production rates were depending on the Photothermal Materials, hybrid of Bi_2Te_3 and Cu showing the best performance. As show in the references listed in this manuscript, the Bi_2Te_3 as photothermal materials have been heavily study. And novelty of hybrid of Bi_2Te_3 and Cu preparation can not reach the level of Nat. Comm.

Response: The authors thank for the reviewer’s comment. As the reviewer said, the Bi_2Te_3 has been heavily studied as photothermal material, but the 1 Sun illuminated temperature of our Bi_2Te_3 structure is $317\text{ }^\circ\text{C}$, far higher than that of any reported Bi_2Te_3 structures (Nano Energy 77 (2020) 105102) and photothermal materials (Nature Energy 6 (2021) 807–814) under 1 Sun illumination. More importantly, this manuscript reveals that the heterostructure of narrow-band gap materials (e. g. Bi_2Te_3) and polished metals (e. g. Cu) can generally improve the sunlight illuminated temperature of narrow-band gap materials, which can provide high-quality heat energy for many fields, such as catalysis, power generation, etc. It is of great innovation and application significance.

Here some comments on the details:

1) In the introduction section: “But, hydrogen generation from methanol and water by

the methanol steam reforming ($\text{CH}_3\text{OH} + \text{H}_2\text{O} = \text{CO}_2 + 3\text{H}_2$, MSR) requires large-scale external energy input (3.2 kJ energy for 1 mol of H_2).^{33,35,36} I did not find this energy number (3.2 kJ energy) in these studies.

Response: The authors thank for the reviewer's valuable question. Firstly, we are sorry for our improper citation, Ref 35, 36 only mentioned MSR. Thus, we have deleted those references and added the correct reference in the revised manuscript.

“33 Sá, S., Silva, H., Brandão, L., Sousa, J. M. & Mendes, A. Catalysts for methanol steam reforming—A review. *Appl. Catal. B: Environ.* **99**, 43-57, (2010).”

Furthermore, when we talk about the external energy input, we use the enthalpy. And ΔH of methanol water reforming is +49.7 kJ/mol. (*Renewable and Sustainable Energy Reviews* 29(2014)355–368), based on this the authors should provide detailed calculation of STH calculation.

Response: The authors thank for the reviewer's valuable comment.

The STH calculation is referred by the published paper of Prof. Domen (*Nature* **581**, 411-414, 2020) and they used ΔG as the chemical energy.

“27 Takata, T. *et al.* Photocatalytic water splitting with a quantum efficiency of almost unity. *Nature* **581**, 411-414, (2020).”

We are very grateful to the reviewer's advice and we have calculated the STH based on ΔH according to the reviewer's suggestion, which was added in the revised manuscript and Supplementary Information.

“Based on the experimental data, the solar-to-hydrogen (STH) conversion efficiency of CuZnAl NS in the $\text{Bi}_2\text{Te}_3/\text{Cu}$ based device was calculated as 31.9 % and 30.1 % under 0.9 and 1 Sun irradiation, respectively (Fig. 5c).”

Fig. 5c The STH efficiency of MSR through CuZnAl NS loaded in the Bi₂Te₃/Cu based device under different intensities of solar irradiation.

“The STH efficiency of hydrogen generation from MSR was calculated as follows:

$$STH = (\Delta H * \epsilon / 24.5) / (I * S * 3600) \quad (1)$$

The enthalpy change energy of methanol, CO₂, H₂, H₂O was -201.083, -393.505, 0, -241.818 kJ mol⁻¹, respectively.

ΔH was the reaction enthalpy change of methanol dehydrogenation ($1/3 \text{ CH}_3\text{OH} (l) + 1/3 \text{ H}_2\text{O} (l) \rightarrow \text{H}_2 (g) + 1/3 \text{ CO}_2 (g)$, $\Delta H = 16.47 \text{ KJ mol}^{-1}$), ϵ (L) was the H₂ generation amount per hour detected by a flowmeter (MV-192-H2), I was the light intensity (kW m⁻²), S was the irradiated area of catalysts (0.0471 m⁻²). The calculation detail and data were shown in Supplementary Fig. 20.”

2) Table 1, the authors should list reaction conditions such as reaction temperature.

Response: Thanks for the reviewer's advice. We have added the reaction temperature in Table 1.

Tab. 1 The sunlight driven hydrogen generation from MSR of CuZnAl NS loaded in Bi₂Te₃/Cu based device, in comparison with the reported advanced sunlight driven hydrogen generation systems. (T: test temperature).

Catalysts	Light source	Reagents	T (°C)	H ₂ rate (mmol·g ⁻¹ ·h ⁻¹)	Ref.
CuZnAl NS	Xe lamp	CH ₃ OH, H ₂ O	25	310	This work
Ni/CdS	Xe lamp	2-propanol, H ₂ O	20	46.6	1
NiS _x /Cd _{0.5} Zn _{0.5} S	Xe lamp	Na ₂ S Na ₂ SO ₃ , H ₂ O	25	44.6	2
Mg-black TiO ₂	Xe lamp	CH ₃ OH, H ₂ O	25	43.1	3
Ni(II)/CdS	Xe lamp	Na ₂ S Na ₂ SO ₃ , H ₂ O	25	43	4
NiO/LaN ₃ TaO ₃	Hg lamp	H ₂ O	25	38.4	5
BP/Bi ₂ WO ₆	Xe lamp	NO CH ₃ OH, H ₂ O	25	21.0	6
C ₃ N ₄ NS	Xe lamp	K ₂ HPO ₄ TEOA,	25	19	7
CdS/2H-MoS ₂	Xe lamp	Na ₂ S Na ₂ SO ₃ , H ₂ O	25	16.6	8
N-doped black TiO ₂	Xe lamp	CH ₃ OH, H ₂ O	25	15.0	9
Sr ₂ MgSi ₂ O ₇ :Eu ²⁺	Hg lamp	CH ₃ OH, H ₂ O	25	14	10
Black TiO ₂	Xe lamp	CH ₃ OH, H ₂ O	25	10	11

3) The authors should list the methanol conversion.

Response: Thanks for the reviewer's good suggestion. We have added the methanol conversion in Supplementary Fig.14 and Supplementary Fig. 16.

Supplementary Fig. 14 Methanol conversion from MSR of CuZnAl NS and C-CuZnAl at different temperatures.

“As a result, the hydrogen production rate of CuZnAl NS was $1.02 \text{ mol g}^{-1} \text{ h}^{-1}$ at $260 \text{ }^\circ\text{C}$, quintupling the $0.2 \text{ mol g}^{-1} \text{ h}^{-1}$ of C-CuZnAl at $260 \text{ }^\circ\text{C}$ (Fig. 4g), and the methanol conversion rate was 5.28 % (Supplementary Fig.14).”

Supplementary Fig. 16 Methanol conversion of CuZnAl NS loaded in the Bi₂Te₃/Cu based device under different sunlight irradiations.

“and the hydrogen generation rate increased to 3.1 mol h⁻¹ (Fig. 5a), under 1 Sun irradiation, corresponding to 310 mmol g⁻¹ h⁻¹ with a methanol conversion rate of 45.34 % (Supplementary Fig. 16).”

4) Methanol water reforming is endothermic reaction, the authors should measure the temperature during the reaction.

Response: Thanks for the reviewer’s very valuable suggestion. We have added the temperature during the sunlight driven reaction in Supplementary Fig. 17.

“Meanwhile, the temperature of the CuZnAl NS loaded in the Bi₂Te₃/Cu based device during solar-heating MSR was shown in Supplementary Fig 17.”

Supplementary Fig. 17 The sunlight driven temperature of the CuZnAl catalyst loaded in the Bi₂Te₃/Cu based device during MSR.

“During the solar heating MSR, the temperature of CuZnAl catalyst loaded in the Bi₂Te₃/Cu based device was 262, 248, 221, 202, 198, 187, 174, 150, 124, 105 °C under 1, 0.9, 0.8, 0.7, 0.6, 0.5, 0.4, 0.3, 0.2, 0.1 Sun irradiation, respectively.”

5) In Figure 1, The thickness of Bi₂Te₃ affects the temperature. I suggest try to measure the temperature of Ti₂O₃/Cu, Cu₂Se/Cu, Cu₂S/Cu with different thickness, then compare the highest results.

Response: Thanks for the reviewer’s good suggestion. We provided the AFM images of different thicknesses of Ti₂O₃/Cu, Cu₂Se/Cu, Cu₂S/Cu and corresponding temperatures, which were shown in supplementary Fig 6 and 7. Those results also confirmed the close relationship between the thickness and sunlight irradiated temperature of photothermal materials.

“Meanwhile, different thicknesses of Ti₂O₃, Cu₂Se, Cu₂S on Cu support were synthesized (Supplementary Fig. 6), and corresponding IR images (Supplementary Fig. 7) showed that the temperature of those samples were all higher than the 1 Sun heating temperature of pure Ti₂O₃ (82 °C), Cu₂Se (79 °C), Cu₂S (75 °C) as shown in Supplementary Fig. 5. Those results confirm that the proper thickness is significant for narrow band semiconductors to have weak IR radiation while maintaining enough solar spectral absorption to create high temperature in the device.”

Supplementary Fig. 6 AFM images of **a** $\text{Ti}_2\text{O}_3/\text{Cu}$ with 17.5 nm thickness of Ti_2O_3 , **b** $\text{Cu}_2\text{Se}/\text{Cu}$ with 13.7 nm thickness of Cu_2Se , **c** $\text{Cu}_2\text{S}/\text{Cu}$ with 26.9 nm thickness of Cu_2S , **d** $\text{Ti}_2\text{O}_3/\text{Cu}$ with 110 nm thickness of Ti_2O_3 , **e** $\text{Cu}_2\text{Se}/\text{Cu}$ with 90 nm thickness of Cu_2Se , **f** $\text{Cu}_2\text{S}/\text{Cu}$ with 135 nm thickness of Cu_2S , **g** $\text{Ti}_2\text{O}_3/\text{Cu}$ with 2.35 μm thickness of Ti_2O_3 , **h** $\text{Cu}_2\text{Se}/\text{Cu}$ with 1.82 μm thickness of Cu_2Se , **i** $\text{Cu}_2\text{S}/\text{Cu}$ with 3.17 μm thickness of Cu_2S .

Supplementary Fig. 7 The corresponding IR images of vacuum protected **a** $\text{Ti}_2\text{O}_3/\text{Cu}$ with 17.5 nm thickness of Ti_2O_3 , **b** $\text{Cu}_2\text{Se}/\text{Cu}$ with 13.7 nm thickness of Cu_2Se , **c** $\text{Cu}_2\text{S}/\text{Cu}$ with 26.9 nm thickness of Cu_2S , **d** $\text{Ti}_2\text{O}_3/\text{Cu}$ with 110 nm thickness of Ti_2O_3 , **e** $\text{Cu}_2\text{Se}/\text{Cu}$ with 90 nm thickness of Cu_2Se , **f** $\text{Cu}_2\text{S}/\text{Cu}$ with 135 nm thickness of Cu_2S , **g** $\text{Ti}_2\text{O}_3/\text{Cu}$ with 2.35 μm thickness of Ti_2O_3 , **h** $\text{Cu}_2\text{Se}/\text{Cu}$ with 1.82 μm thickness of Cu_2Se , **i** $\text{Cu}_2\text{S}/\text{Cu}$ with 3.17 μm thickness of Cu_2S , under 1 Sun irradiation. A CaF_2 glass fully covered the materials, and the vacuum degree of this equipment was 1.0×10^{-3} Pa.

“The deposition of Ti_2O_3 , Cu_2Se , Cu_2S films

The synthesis of $\text{Ti}_2\text{O}_3/\text{Cu}$, $\text{Cu}_2\text{Se}/\text{Cu}$, $\text{Cu}_2\text{S}/\text{Cu}$ shown in Supplementary Fig. 4a, Supplementary Fig. 4b, Supplementary Fig. 4c, Supplementary Fig. 6a, Supplementary Fig. 6b, Supplementary Fig. 6c, Supplementary Fig. 6d, Supplementary Fig. 6e, Supplementary Fig. 6f was similar to that of $\text{Bi}_2\text{Te}_3/\text{Cu}$ with 100 nm thickness of Bi_2Te_3 . And the only difference was changing the Bi_2Te_3 target by Ti_2O_3 target, Cu_2Se target or Cu_2S target. The sputtering time of films shown in Supplementary Fig. 4a, Supplementary Fig. 4b, Supplementary Fig. 4c, Supplementary Fig. 6a, Supplementary Fig. 6b, Supplementary Fig. 6c, Supplementary Fig. 6d, Supplementary Fig. 6e, Supplementary Fig. 6f was 6 min, 6 min, 6 min, 1 min, 1 min, 1 min, 3 min, 3 min, 3 min, respectively.

For the films shown in Supplementary Fig. 6g, Supplementary Fig. 6h, Supplementary Fig. 6i, the sputtering pressure was 5×10^{-1} Pa and the sputtering time was 10 min.”

6) The Figure 2 d is confusing.

Response: Thanks for the reviewer’s question. We are sorry for our confusing diagram. To avoid confusing, we have deleted the Figure 2d in manuscript.

7) Figure 4b is not meaningful. I suggest the authors label the scale bars in Figure c d e.

Response: Thanks for the reviewer’s good suggestion, we have changed those figures according to the reviewer’s advice in the revised manuscript and Supplementary Information.

Fig. 4 a Schematic of the synthesis process for CuZnAl NS. The TEM image b, elemental mapping c, HRTEM image d, AFM image e, nitrogen adsorption and

desorption isotherm **f** of CuZnAl NS. **g** Hydrogen production rate from MSR of CuZnAl NS and C-CuZnAl at different temperatures. Test condition: 10 mg of catalyst, 50 sccm of Ar, 0.1 sccm of methanol (the volume ratio of methanol to water=1:1.3). The scale bars in **b**, **c**, **d** are 200, 500, 2 nm, respectively.

Supplementary Fig. 13 Photograph of the bottle full of as-synthesized CuZnAl NS.

“Supplementary Fig. 13 showed that our synthesized sample could full fill a 40 L bottle, revealing the ability of scalable preparation.”

“As a result, the hydrogen production rate of CuZnAl NS was $1.02 \text{ mol g}^{-1} \text{ h}^{-1}$ at $260 \text{ }^{\circ}\text{C}$, quintupling the $0.2 \text{ mol g}^{-1} \text{ h}^{-1}$ of C-CuZnAl at $260 \text{ }^{\circ}\text{C}$ (Fig. 4g), and the methanol conversion rate was 5.28 % (Supplementary Fig.14).”

Reviewer #2 (Remarks to the Author):

The work by Li, Wang, Luo, and coworkers demonstrate a novel photothermal material for converting sunlight into non-artificial heat with unprecedented high temperatures reached. The authors demonstrate the use of this material by performing methanol steam reforming (MSR) at 260 degrees Celcius entirely driven by solar heating, which is very impressive and highly interesting in my mind.

However, solar photothermal driven thermochemical MSR reactions is not new, as the authors point out themselves. Nevertheless, the presented work does provide an improvement of approximately an order of magnitude compared to the current state-of-the-art, and their device is stable for more than 20 days. I find it confusing, though, that the authors apparently go on to synthesize a new MSR catalyst to confirm the use of their novel photothermal material, instead of demonstrating it with already published thermally driven catalytic reactions. How does their approach validate the superiority of Bi₂Te₃/Cu over other solar-to-thermal conversion devices? And how do we really compare the results when both the catalyst and the photothermal device are new? If I am to assess the overall activity and stability of the MSR, the work is, in my view, of high significance, but does not warrant publication in Nature Communications in its current form. However, the work should be re-assessed after the following considerations:

Response: We are grateful for the reviewer's comprehensive review.

This ultra-high sunlight driven MSR performance must be the synergistic effect of the new device and the new catalyst. As the reviewer mentioned, we should explain the individual effect of the new device and new catalyst. Therefore, we added a comparative test, that is, we added a commercial CuZnAl (C-CuZnAl) catalyst to the Bi₂Te₃/Cu based device and carried out the photothermal experiment on it. We found that the 1 Sun driven MSR H₂ generation rate was 79.3 mmol g⁻¹ h⁻¹ (supplementary Fig 11), which was much higher than all reported record photocatalytic MSR (46.6 mmol g⁻¹ h⁻¹), thus highlighting the importance of the Bi₂Te₃/Cu based device. At the

same time, in order to obtain higher sunlight driven H₂ generation rate, we prepared the new catalyst (CuZnAl nanosheets) and loaded it into the Bi₂Te₃/Cu based device. The result shows that the joint can further enhance the 1 Sun driven MSR H₂ generation rate to 310 mmol g⁻¹ h⁻¹. Therefore, the ultra-high sunlight driven MSR H₂ generation rate is the synergistic effect of the Bi₂Te₃/Cu based device and the CuZnAl nanosheets. We have added the data in supplementary Fig 11 and added some descriptions in the revised manuscript.

Supplementary Fig. 11 The hydrogen production rates from two MSRs of C-CuZnAl loaded in the Bi₂Te₃/Cu based device and C-CuZnAl not in the device but directly under different sunlight irradiations.

“we added the commercial CuZnAl (C-CuZnAl, Supplementary Fig. 10) into Bi₂Te₃/Cu based device to test the sunlight driven MSR performance. As shown in Supplementary Fig. 11, the 1 Sun driven MSR H₂ generation rate through C-CuZnAl was 79.3 mmol g⁻¹ h⁻¹, which is much higher than that reported record photocatalytic MSR value (46.6 mmol g⁻¹ h⁻¹),¹ thus highlighting the importance of the Bi₂Te₃/Cu based device. To achieve higher sunlight driven MSR performance, the more efficient catalysts for MSR need to be developed.”

1) The concept of “Methanol economy” by Olah ought to be mentioned.

Response: We thanks for the reviewer’s good suggestion. Therefore, we have added the “Methanol economy” by Olah in the revised manuscript.

“Owing to those storage limitations of hydrogen such as high pressure, leakage, and extensive safety precautions, Olah has proposed methanol economy as the methanol can be act as the hydrogen carrier in the future hydrogen energy system,^{12”}

“29 Olah, G. A. Beyond Oil and Gas: The Methanol Economy. *Angew. Chem. Int. Ed.* **44**, 2636-2639, (2005).”

2) If the photothermal device is merely developed to provide breakthrough-level performing solar driven MSR, I suggest to further optimize the results to obtain a more significant improvement over state-of-the-art. If the MSR is included to merely highlighting the power of the photothermal device, I suggest to test and compare outcome when using known catalysts for MSR.

Response: Thanks for the reviewer’s good suggestion. We firstly want to highlight the power of the photothermal device. Therefore, we added the benchmark MSR catalyst: commercial CuZnAl into the Bi₂Te₃/Cu based device and the 1 Sun driven MSR H₂ generation rate was 79.3 mmol g⁻¹ h⁻¹ (supplementary Fig 11), which was much higher than that reported record photocatalytic MSR value (46.6 mmol g⁻¹ h⁻¹), thus highlighting the importance of the Bi₂Te₃/Cu based device. Secondly, we also want to show the high activity of the new catalyst for MSR. We prepared a new catalyst (CuZnAl nanosheets), exhibiting five times MSR activity compared with commercial CuZnAl, as we loaded CuZnAl nanosheets into the Bi₂Te₃/Cu based device. The result shows that the joint can further enhance the 1 Sun driven MSR H₂ generation rate to 310 mmol g⁻¹ h⁻¹. Therefore, the ultra-high sunlight driven MSR H₂ generation rate is the synergistic effect of the Bi₂Te₃/Cu based device and the CuZnAl nanosheets. We have added the data in supplementary Fig 11 and added some descriptions in the revised manuscript.

Supplementary Fig. 11 The hydrogen production rates from two MSRs of C-CuZnAl loaded in the Bi₂Te₃/Cu based device and C-CuZnAl not in the device but directly under different sunlight irradiations.

“we added the commercial CuZnAl (C-CuZnAl, Supplementary Fig. 10) into Bi₂Te₃/Cu based device to test the sunlight driven MSR performance. As shown in Supplementary Fig. 11, the 1 Sun driven MSR H₂ generation rate through C-CuZnAl was 79.3 mmol g⁻¹ h⁻¹, which is much higher than that reported record photocatalytic MSR value (46.6 mmol g⁻¹ h⁻¹),¹ thus highlighting the importance of the Bi₂Te₃/Cu based device. To achieve higher sunlight driven MSR performance, the more efficient catalysts for MSR need to be developed.”

3) The authors employ MSR as a benchmarking reaction for conducting a thermally driven reaction at a high temperature above 120 degrees Celcius using their novel photothermal device. In my mind, it would be pertinent at least to acknowledge that there indeed exist low-temperature methanol reforming protocols performing well below 120 degrees Celcius (e.g. Beller Nature 2013, 495, 85 as well as Trincado and Grützmaier Nature Chemistry 2013, 5, 342), and to explain why it is pertinent to develop a sun-to-heat device for high-temperature MSR when low-temperature systems

have already been developed on for almost a decade. An explanation becomes even more relevant when considering that PEM fuel cells generate an excess of heat that the low-temperature MSR potentially could exploit.

Response: Thanks for the reviewer's good question. The reviewer listed papers are great works and we have referred them in the revised manuscript. The catalysts, e.g. $[\text{RuHCl}(\text{CO})(\text{HN}(\text{C}_2\text{H}_4\text{PiPr}_2)_2)]$,¹³ $[\text{K}(\text{dme})_2][\text{Ru}(\text{H})(\text{trop}_2\text{dad})]$ ¹⁴ can drive MSR at very low temperature, which are the most suitable catalysts for solar heating MSR. Unfortunately, for our work, we want to do a large scale demo of solar heating MSR, which requires at least kg level catalysts. However, we cannot synthesize this type of excellent catalysts e.g., ($[\text{RuHCl}(\text{CO})(\text{HN}(\text{C}_2\text{H}_4\text{PiPr}_2)_2]$, $[\text{K}(\text{dme})_2][\text{Ru}(\text{H})(\text{trop}_2\text{dad})]$) even in 10 g level, due to the high prices and rare resources of Ru. Therefore, we selected the CuZnAl catalyst as the benchmark MSR catalyst, because of the merits of low cost, high abundance and large scale production. Our future work is to synthesize low-temperature MSR catalysts, with the advantages of mass synthesis, low cost and high abundance, which is a challenging research field.

“40 Nielsen, M. *et al.* Low-temperature aqueous-phase methanol dehydrogenation to hydrogen and carbon dioxide. *Nature* **495**, 85-89, (2013).

41 Rodríguez-Lugo, R. E. *et al.* A homogeneous transition metal complex for clean hydrogen production from methanol–water mixtures. *Nat. Chem.* **5**, 342-347, (2013). “

4) The authors mention that their system produces a low concentration of CO; however, 0.8% of CO (of CO₂ and CO) is not a low concentration for PEM fuel cells. Please consider this more carefully, especially in the light of that they mention how this system may be used for powering a vehicle.

Response: Thanks for the reviewer's comment. The CO concentration that can be used for PEM fuel cells is less than 10 ppm (Beller *Nature* 2013, 495, 85), the hydrogen produced by our system requires purification for a vehicle that we will do in the future.

Therefore, we have deleted the expression in the manuscript.

5) It would be instrumental with some more pictures that clearly show the function and setup of the device, eg of the outdoor setup.

Response: Thanks for the reviewer's comment. We provide a video (Supplementary Movie 1) for the outdoor setup to facilitate reviewers' reading.

Reviewer #3 (Remarks to the Author):

In this manuscript, the authors propose to combine black photothermal materials and infrared insulating materials, so that an elevated solar-heating temperature can be achieved. As an example, the Bi₂Te₃/Cu hybrid showed a 1 Sun-heating temperature of 317 °C. In addition, a reaction device based on Bi₂Te₃/Cu was assembled, which was able to heat a catalyst of CuO_x/ZnO/Al₂O₃ nanosheets to 305 °C and showed a 1 Sun driven hydrogen production rate of 75.9 L/h from methanol and water. These results are impressive, and the following issues need addressed before possible publication of this manuscript.

Response: The authors are grateful for the reviewer's thorough comment.

1 When the thickness of Bi₂Te₃ layer was 100 nm, the Bi₂Te₃/Cu exhibited the highest 1 Sun irradiated temperature of 317 °C, higher than those for 15nm and 3 micron thick Bi₂Te₃. Is this an optimized result? What is the reason behind?

Response: Thanks for the reviewer's good question. The 100 nm thickness of Bi₂Te₃ layer was an optimized result. The physical reason is the synergistic effect of heat radiation and sunlight absorption of Bi₂Te₃ layer. The IR radiation from Bi₂Te₃ is produced by lattice vibrations, proportional to the number of atoms in Bi₂Te₃ film, in other words, the thinner the Bi₂Te₃ film is, the less the intensity of IR radiation is. For comparison, we found that the sunlight absorption of Bi₂Te₃ film was significantly decreased from 89 % to 43 % as the thickness of Bi₂Te₃ film decreased from 100 nm to 15 nm. Therefore, the 100 nm thickness of Bi₂Te₃ layer can synergistically absorb 89 % of sunlight and emit 5 % of IR radiation, in other words, this 100 nm thickness of Bi₂Te₃ layer can absorb sunlight energy to the maximum extent and dissipate heat energy to the minimum extent, so that the heat energy converted from sunlight is localized in the interior of the device, resulting in high temperature. We have made a table (Supplementary Tab.1) to show the solar absorptivity and infrared emissivity of Bi₂Te₃ layers with different thicknesses more clearly.

Supplementary Tab. 1 The solar absorptivity, IR emissivity, 1 Sun irradiated temperature of Bi₂Te₃/Cu film with different thickness of Bi₂Te₃. A CaF₂ glass fully covered each sample, and the vacuum degree of this equipment was 1.0×10⁻³ Pa.

Thickness	Solar absorptivity	IR emissivity	1 Sun irradiated temperature
15 nm	43 %	4 %	172 °C
100 nm	89 %	5 %	317 °C
3 μm	94 %	60 %	97 °C

“Summing up the solar absorptivity and IR emissivity listed in Supplementary Tab.1, the 100 nm thickness of Bi₂Te₃ layer can synergistically absorb 89 % of sunlight and emit 5 % of IR radiation, in other words, this 100 nm thickness of Bi₂Te₃ layer can absorb sunlight energy to the maximum extent and dissipate heat energy to the minimum extent, so that the heat energy converted from sunlight is localized in the interior of the Bi₂Te₃ layer, resulting in high sunlight irradiated temperature.”

2 What is the fine structure of the Bi₂Te₃ layer? As seen from the TEM images, there seems to be difference between the structures of the Bi₂Te₃ layer with different thicknesses. Is the heating performance of the hybrid only thickness-related, or influenced by the structure of Bi₂Te₃?

Response: Thanks for the reviewer’s question. We are also very confused about this phenomenon. After discussion with experts, we think that the morphology difference of samples might be caused by the sample preparation. To check this speculation, TEM samples of Bi₂Te₃/Cu films with 100 nm and 15 nm thickness of Bi₂Te₃ were prepared by ion thinning, and then it was found that the cross-section TEM images of the two samples were similar (Fig. 1d, e). In comparison, the SEM sample of Bi₂Te₃/Cu film with 3 μm thickness of Bi₂Te₃ (Fig. 1c) is synthesized not by ion thinning but by direct shear. Therefore, we can see that the cross-section SEM image is rough, different from

the TEM images.

Fig. 1 c, d, e SEM and TEM images of Bi₂Te₃/Cu with 3 μm, 100 nm, 15 nm thickness of Bi₂Te₃. The scale bars in c, d, e are 1500, 30, 5 nm, respectively.

3 What is the Bi₂Te₃/Cu interface structure? How does it affect the Sun-heating performance?

Response: The authors thank for the reviewer's question. We provided the interface structure of Bi₂Te₃/Cu in Supplementary Fig. 2. The description of interface structure and effect on solar heating catalysis is added in Supplementary Information, which is also shown below here for the convenience of the reviewer.

Supplementary Fig. 2 The HRTEM image of the interface of Bi_2Te_3 and Cu.

“Supplementary Fig. 2 clearly showed that the crystal planes of Cu (111), CuO (002), Bi_2Te_3 (015) grown in a similar direction. This layer of CuO may be caused by the intrinsic oxidation of the Cu substrate in air. The sunlight can be efficiently absorbed by the Bi_2Te_3 layer to create a high temperature, and heat energy flow is smoothly transferred from Bi_2Te_3 layer to Cu layer relied on the interface structure, thus heating the device and catalysts efficiently.”

4 An illustration and corresponding discussion clearly showing the working mechanism of this hybrid should be presented, i.e. how is the sun light irradiation transferred to thermal energy efficiently through the Bi_2Te_3 layer, Cu and their interface? What is the

key physics of this hybrid design?

Response: The authors thank for the reviewer's comment and we have added the description in Supplementary Fig. 8, which is also shown below here for the convenience of the reviewer.

Supplementary Fig. 8 The picture of catalysts loaded in Bi₂Te₃/Cu based device.

“When sunlight directly illuminated the solar heating device, the layer of Bi₂Te₃ could efficiently convert solar energy to thermal energy and radiate little infrared light; the inner layer of Cu could stop infrared emission of catalysts; while the outer vacuum layer can block the thermal conduction from the inner device. In this way, we could create a high temperature to drive MSR under solar illumination (Supplementary Fig. 8).

The key physics of this hybrid is that the infrared radiation is directly proportional to the amount of sunlight absorber (Bi₂Te₃). Therefore, the thinner the thickness of Bi₂Te₃, the less its heat radiation will be. At the same time, if the thickness of Bi₂Te₃ film is less

than 100 nm, its ability to absorb sunlight will be greatly reduced. Therefore, controlling the thickness of Bi₂Te₃ to about 100 nm can achieve the balance of high sunlight absorption and low infrared radiation to produce high sunlight irradiation temperature.”

References

- 1 Chai, Z. *et al.* Efficient Visible Light-Driven Splitting of Alcohols into Hydrogen and Corresponding Carbonyl Compounds over a Ni-Modified CdS Photocatalyst. *J. Am. Chem. Soc.* **138**, 10128-10131, (2016).
- 2 Liu, M. *et al.* Photocatalytic hydrogen production using twinned nanocrystals and an unanchored NiS_x co-catalyst. *Nat. Energy* **1**, 16151, (2016).
- 3 Sinhamahapatra, A., Jeon, J.-P. & Yu, J.-S. A new approach to prepare highly active and stable black titania for visible light-assisted hydrogen production. *Energy Environ. Sci.* **8**, 3539-3544, (2015).
- 4 Zhao, G. *et al.* Superior Photocatalytic H₂ Production with Cocatalytic Co/Ni Species Anchored on Sulfide Semiconductor. *Adv. Mater.* **29**, 1703258, (2017).
- 5 Kato, H., Asakura, K. & Kudo, A. Highly Efficient Water Splitting into H₂ and O₂ over Lanthanum-Doped NaTaO₃ Photocatalysts with High Crystallinity and Surface Nanostructure. *J. Am. Chem. Soc.* **125**, 3082-3089, (2003).
- 6 Hu, J. *et al.* Z-Scheme 2D/2D Heterojunction of Black Phosphorus/Monolayer Bi₂WO₆ Nanosheets with Enhanced Photocatalytic Activities. *Angew. Chem. Int. Edit.* **58**, 2073-2077, (2019).
- 7 Liu, G. *et al.* Nature-Inspired Environmental “Phosphorylation” Boosts Photocatalytic H₂ Production over Carbon Nitride Nanosheets under Visible-Light Irradiation. *Angew. Chem. Int. Edit.* **54**, 13561-13565, (2015).
- 8 Chang, K. *et al.* Targeted Synthesis of 2H- and 1T-Phase MoS₂ Monolayers for Catalytic Hydrogen Evolution. *Adv. Mater.* **28**, 10033-10041, (2016).
- 9 Lin, T. *et al.* Effective nonmetal incorporation in black titania with enhanced solar energy utilization. *Energy Environ. Sci.* **7**, 967, (2014).
- 10 Cui, G. *et al.* Round-the-Clock Photocatalytic Hydrogen Production with High Efficiency by a Long-Afterglow Material. *Angew. Chem. Int. Edit.* **58**, 1340-1344, (2019).
- 11 Chen, X., Liu, L., Yu, P. Y. & Mao, S. S. Increasing Solar Absorption for Photocatalysis with Black Hydrogenated Titanium Dioxide Nanocrystals. *Science* **331**, 746-750, (2011).
- 12 Olah, G. A. Beyond Oil and Gas: The Methanol Economy. *Angew. Chem. Int. Edit.* **44**, 2636-2639, (2005).
- 13 Nielsen, M. *et al.* Low-temperature aqueous-phase methanol dehydrogenation to hydrogen and carbon dioxide. *Nature* **495**, 85-89, (2013).
- 14 Rodríguez-Lugo, R. E. *et al.* A homogeneous transition metal complex for clean hydrogen production from methanol–water mixtures. *Nat. Chem.* **5**, 342-347,

(2013).

REVIEWER COMMENTS

Reviewer #1 (Remarks to the Author):

Dear Editor and authors,

I carefully read through the revised manuscript and response letter. The authors made a perfect revision, all my concern have been addressed.

In the first round of review, the only reason I did not recommend publishing is that this study is engineering, or device orientated. I found some similar studies have been published in the Nature Communications previously. More importantly, the authors convinced me with their careful and perfect revision.

Therefore, I recommend publishing it in Nature Communications.

Reviewer #2 (Remarks to the Author):

In this resubmission by Li et al, the authors fully address my concerns regarding their initial submission. However, in my mind, the outcome of this effort does not lead to a satisfactory improvement of the overall work for warranting publication in Nature Communications. More specifically, the reported catalytic performance of the MSR is indeed significantly superior to literature precedence, but they are not radical improvements. As such, I recommend publishing in a more specialized journal.

Reviewer #3 (Remarks to the Author):

The authors answered my questions, but not completely. My main concern is how the Bi₃Te₂/Cu system achieves a superior solar-to-thermal conversion performance, which is a main novelty of this work. Is it related to the thickness and structure of Bi₃Te₂ layer, the Bi₃Te₂/Cu interface, or some new physics of this device? However, the authors did not give a clear answer and description on this point. As a result, there is insufficient deepened discussion and understanding on the achieved device and phenomena. At the current stage, this manuscript is not suitable for publication in Nature Commun.

REVIEWER COMMENTS

Reviewer #1 (Remarks to the Author):

Dear Editor and authors,

I carefully read through the revised manuscript and response letter. The authors made a perfect revision, all my concern have been addressed.

In the first round of review, the only reason I did not recommend publishing is that this study is engineering, or device orientated. I found some similar studies have been published in the Nature Communications previously. More importantly, the authors convinced me with their careful and perfect revision.

Therefore, I recommend publishing it in Nature Communications.

Response: The authors very thank indeed for your professional and kind support. Through your guidance, we have a deeper understanding of this novel field and have greatly improved our manuscript.

Reviewer #2 (Remarks to the Author):

In this resubmission by Li et al, the authors fully address my concerns regarding their initial submission. However, in my mind, the outcome of this effort does not lead to a satisfactory improvement of the overall work for warranting publication in Nature Communications. More specifically, the reported catalytic performance of the MSR is

indeed significantly superior to literature precedence, but they are not radical improvements. As such, I recommend publishing in a more specialized journal.

Response: The authors thank for your helpful comment on our catalytic performance of the MSR that is indeed significantly superior to literature precedence. We also believe that the performance is indeed a radical improvement. This is because, firstly, the 1 Sun illuminated temperature of our Bi₂Te₃ structure is 317 °C, which is not only 224 °C higher than that of pure Bi₂Te₃ (93 °C) but also 197 °C higher than the reported highest temperature of photothermal materials (120 °C) under 1 Sun irradiation (Adv. Funct. Mater. 31, 7, 2021).¹ More importantly, the catalytic hydrogen production performance of the solar heating MSR by our system is revolutionary, as it is far higher than the theoretical limit of photocatalytic MSR, for which detailed data, descriptions and references are presented below for your convenience. They have also been added in the revised manuscript and Supplementary Information.

“We need to point out that the STH of our solar heating MSR is beyond the theoretical limit of STH of photocatalytic MSR through the route of photons-photogenerated electrons and holes-chemicals.² Ishii et al. reported that the average energy of photons in solar spectrum is ~ 1 eV.³ However, the reaction enthalpy of MSR is 0.086 eV_{per H} (1/3 CH₃OH (l) + 1/3 H₂O (l) → H₂ (g) + 1/3 CO₂ (g), Detailed calculation seen in Methods). Therefore, the STH ceiling of photocatalytic MSR under 1 Sun illumination is 8.6% (0.086 eV/1 eV), equivalent to ~1/3 of the STH of our solar heating MSR (30.1%) under 1 Sun irradiation. This work reveals that solar heating catalysis via solar-thermal energy-chemicals route has an incomparable advantage compared with

photocatalysis in the reactions with low energy barrier.”

“As $1 \text{ eV} = 1.6 \times 10^{-19} \text{ J}$, the ΔH for per H_2 was calculated as $16.47 \text{ KJ}/(1.6 \times 10^{-19} * 6.02 \times 10^{23}) = 0.171 \text{ eV}$, therefore, the ΔH for per H was calculated as $0.171 \text{ eV}/2 = 0.086 \text{ eV}$.”

References

- 1 Han, X. M. *et al.* Intensifying Heat Using MOF-Isolated Graphene for Solar-Driven Seawater Desalination at 98% Solar-to-Thermal Efficiency. *Adv. Funct. Mater.* **31**, 7, (2021).
- 2 Nishiyama, H. *et al.* Photocatalytic solar hydrogen production from water on a 100 m² scale. *Nature* **598**, 304-307, (2021).
- 3 Ishii, T., Otani, K., Takashima, T. & Xue, Y. Solar spectral influence on the performance of photovoltaic (PV) modules under fine weather and cloudy weather conditions. *Prog. Photovoltaics Res. Appl.* **21**, 481-489, (2013).”

Reviewer #3 (Remarks to the Author):

The authors answered my questions, but not completely. My main concern is how the $\text{Bi}_2\text{Te}_3/\text{Cu}$ system achieves a superior solar-to-thermal conversion performance, which is a main novelty of this work. Is it related to the thickness and structure of $\text{Bi}_2\text{Te}_3/\text{Cu}$ layer, the $\text{Bi}_2\text{Te}_3/\text{Cu}$ interface, or some new physics of this device? However, the authors did not give a clear answer and description on this point. As a result, there is insufficient deepened discussion and understanding on the achieved device and

phenomena. At the current stage, this manuscript is not suitable for publication in Nature Commun.

Response: The authors thank for your helpful comment, and we are very sorry for misunderstanding your previous comment. We have added new data and descriptions for explaining the superior solar-to-thermal conversion performance of $\text{Bi}_2\text{Te}_3/\text{Cu}$ and highlighting the key innovation of our work. The details are categorized into 4 parts about the roles of Bi_2Te_3 , the $\text{Bi}_2\text{Te}_3/\text{Cu}$ heterostructure, the thickness effect, and the $\text{Bi}_2\text{Te}_3/\text{Cu}$ interface. They are presented below for your convenience, indicating that the superior solar-to-thermal conversion performance originates from the material nature of Bi_2Te_3 , the low IR radiation is caused by the combination of $\text{Bi}_2\text{Te}_3/\text{Cu}$ heterostructure and the thickness control of the Bi_2Te_3 thin film, the excellent heat transport is caused by the $\text{Bi}_2\text{Te}_3/\text{Cu}$ interface.

1. Bi_2Te_3 is a typical material with narrow bandgap of < 0.2 eV to be able to fully absorb sunlight and convert it as heat energy, therefore, the most important factor to cause the superior solar-to-thermal conversion performance of $\text{Bi}_2\text{Te}_3/\text{Cu}$ is the material nature of Bi_2Te_3 , which has been confirmed by many reported and our studies (the details are presented in the following data, descriptions and references, which have also been added in the revised manuscript and Supplementary Information).

Supplementary Fig. 1 a, b Photograph and normalized light absorption spectra of pure Bi₂Te₃ film. **c** The carrier concentration of pure Bi₂Te₃ film under different temperature.

“Bi₂Te₃ is a typical photothermal material with narrow-band gap (<0.2 eV)^{4,5} to nearly full absorb solar spectrum (Supplementary Fig. 1a, b) and has high carrier concentration of $0.84\text{-}1.11 \times 10^{19} \text{ cm}^{-3}$ (Supplementary Fig. 1c). Therefore, the absorbed sunlight can be fully thermalized by this type of narrow-bandgap semiconductor via electron-phonon and electron-electron scattering.⁶ For instance, Cheng et al. reported that Bi₂Te₃ could convert maximum of 99% solar energy as heat energy.⁷”

References

- 4 Chen, Y. L. *et al.* Experimental Realization of a Three-Dimensional Topological Insulator Bi₂Te₃. *Science* **325**, 178-181, (2009).
- 5 Dheepa, J., Sathyamoorthy, R. & Subbarayan, A. Optical properties of thermally evaporated Bi₂Te₃ thin films. *J. Cryst. Growth* **274**, 100-105, (2005).
- 6 Wang, J. *et al.* High-Performance Photothermal Conversion of Narrow-Bandgap Ti₂O₃ Nanoparticles. *Adv. Mater.* **29**, 1603730, (2017).
- 7 Wang, Z., Zhang, Z. M., Quan, X. & Cheng, P. A perfect absorber design using a natural hyperbolic material for harvesting solar energy. *Solar Energy* **159**, 329-336,

(2018).

2. On the basis that Bi_2Te_3 can naturally saturately absorb sunlight and convert it into heat energy, one of the main innovations of our work is to use the $\text{Bi}_2\text{Te}_3/\text{Cu}$ heterostructure to reduce IR radiation of Bi_2Te_3 to greatly boost the solar irradiated temperature, whose detailed mechanism is presented in the following data and description that have also been added in the revised manuscript and Supplementary Information.

Supplementary Fig. 1d IR image of vacuum protected pure Bi_2Te_3 film under 1 Sun irradiation. A CaF_2 glass fully covered the samples, and the vacuum degree of this equipment was 1.0×10^{-3} Pa.

Fig. 1 a Sketch map of heat dissipation of pure Bi_2Te_3 film. b Synthesis sketch map of

Bi₂Te₃ thin film on Cu support (Bi₂Te₃/Cu).

“For achieving high solar irradiated temperature, in addition to superior solar-to-thermal conversion, it is also necessary to localize the sunlight converted heat energy in Bi₂Te₃, that is reducing the heat dissipation of Bi₂Te₃. Despite a vacuum protection was applied to cut off the heat conduction loss of pure Bi₂Te₃ film, the 1 Sun (1 kW m⁻²) illuminated temperature of pure Bi₂Te₃ film was only 93 °C (Supplementary Fig. 1d). As a blackbody material (Fig. 1a),⁸ the heat dissipation of pure Bi₂Te₃ film includes not only the heat conduction loss but also importantly the violent heat radiation loss caused by the infrared light (IR) radiation (0.91 of IR emissivity shown in Supplementary Information).⁹ Therefore, minimizing the IR radiation of Bi₂Te₃ is the key for increasing its solar irradiated temperature. The IR lights radiated by Bi₂Te₃ are produced by lattice vibrations and the lattice vibrations are proportional to the number of atoms in Bi₂Te₃.¹⁰ From the physical principle, reducing the number of atoms of Bi₂Te₃ structure can weaken the IR radiation, so that our strategy is synthesizing Bi₂Te₃ thin film to minimize the number of atoms to minimize the IR radiation as shown in Fig. 1b. In order to achieve low IR radiation of Bi₂Te₃ thin film structure, the supports used for depositing Bi₂Te₃ thin film need to have the property of low IR radiation too. However, the supports used for depositing Bi₂Te₃ thin film are usually silicon, which is also a typical blackbody material with strong IR radiation and cannot be used as the support for reducing the IR radiation of whole Bi₂Te₃ thin film structure.⁸ Different from blackbody materials, highly conductive metal of Cu contains a large number of near free electrons that can prevent the spillover of IR lights,¹¹ and makes Cu have near zero

IR radiation (~3 % of IR emissivity, Supplementary Fig. 2).^{12,13} Therefore, Cu film is selected as the support for synthesizing Bi₂Te₃ thin film to make the hybrid have the merits of superior solar-to-thermal conversion from Bi₂Te₃ and low IR radiation from Cu.¹⁴

References

- 8 Li, X. J. *et al.* Bi₂Te₃/Si Thermophotovoltaic Cells Converting Low-Temperature Radiation into Electricity. *Phys. Rev. Appl.* **13**, (2020).
- 9 Ni, G. *et al.* Steam generation under one sun enabled by a floating structure with thermal concentration. *Nat. Energy* **1**, 16126, (2016).
- 10 Rice, A. M. *et al.* Photophysics Modulation in Photoswitchable Metal-Organic Frameworks. *Chem. Rev.* **120**, 8790-8813, (2020).
- 11 Ning, Y. P. *et al.* NiCrAlO/Al₂O₃ solar selective coating prepared by direct current magnetron sputtering and water boiling. *Sol. Energy Mater. Sol. Cells* **219**, 6, (2021).
- 12 Gong, S. *et al.* Preparation of ATO-incorporated composite latex with tailored structure and controllable size for highly spectrum-selective applications. *Mater. Des.* **180**, 12, (2019).
- 13 Dao, T. D. *et al.* Infrared Perfect Absorbers Fabricated by Colloidal Mask Etching of Al–Al₂O₃–Al Trilayers. *ACS Photonics* **2**, 964-970, (2015).
- 14 Zhang, J. *et al.* A flexible film to block solar radiation for daytime radiative cooling. *Sol. Energy Mater. Sol. Cells* **225**, 12, (2021).

3. The 1 Sun illuminated temperature of the Bi₂Te₃/Cu heterostructure can be as high

as 317 °C through controlling the thickness of the Bi₂Te₃ thin film in Bi₂Te₃/Cu, which is not only 224 °C higher than that of pure Bi₂Te₃ (93 °C) but also 197 °C higher than the reported highest temperature of photothermal materials (120 °C) under 1 Sun irradiation (Adv. Funct. Mater. 31, 7, 2021).¹ The mechanism of the Bi₂Te₃ thickness control is to balance the solar absorption and the IR radiation of Bi₂Te₃/Cu. We have described the detail in the revised manuscript and also place the description below for your convenience. Further, in order to show more clearly the thickness effect of the Bi₂Te₃/Cu heterostructure on the IR radiation, we have added the IR radiation intensities of all samples at 93 °C in Fig. 2d, e, f and Supplementary Fig. 1e.

Fig. 2 a, b, c Normalized light absorption spectra of Bi₂Te₃/Cu with 3 μm, 100 nm, 15 nm thickness of Bi₂Te₃ layer. **d, e, f** The IR radiation ranging from 4 μm to 20 μm of the Bi₂Te₃/Cu with 3 μm, 100 nm, 15 nm thickness of Bi₂Te₃ layer at 93 °C.

Supplementary Fig. 1e The IR radiation ranging from 4 μm to 20 μm of the pure Bi_2Te_3 film at 93 $^\circ\text{C}$.

“In order to explain the Bi_2Te_3 thin film thickness effect on the sunlight irradiated temperature of $\text{Bi}_2\text{Te}_3/\text{Cu}$, we measured the light absorption of the three $\text{Bi}_2\text{Te}_3/\text{Cu}$ samples. For the 3 μm , 100 nm and 15 nm thicknesses of Bi_2Te_3 thin film in $\text{Bi}_2\text{Te}_3/\text{Cu}$, Fig. 2a, b, c showed the absorbance in sunlight region (400 nm-2000 nm) of $\sim 94\%$, 89%, 43%, respectively. Bi_2Te_3 has a narrow bandgap of $< 0.2\text{ eV}$,^{4,5} thus, the sunlight has enough energy to excite electrons transition in Bi_2Te_3 .^{15,16} But, the film thickness of Bi_2Te_3 must be $\geq 100\text{ nm}$ to ensure more than 89% of solar spectrum absorption. Whereas, the IR region absorption was 4%, 5% as the thickness of Bi_2Te_3 thin film in $\text{Bi}_2\text{Te}_3/\text{Cu}$ was 15, 100 nm, respectively (Fig. 2b, c), and it increased to 60% when the thickness of Bi_2Te_3 thin film extended to 3 μm (Fig. 2a). As the absorptivity of light is equal to the emissivity of corresponding light,¹¹ the 60% IR absorption depicted that the IR emissivity of $\text{Bi}_2\text{Te}_3/\text{Cu}$ with 3 μm of Bi_2Te_3 thin film is 60%, at least 10 times higher than the $\text{Bi}_2\text{Te}_3/\text{Cu}$ with 100 nm (5%), 15 nm (4%) thickness of Bi_2Te_3 thin

film. For a more intuitive embodiment, we directly tested the IR radiation intensity (4 μm -20 μm) of these samples heated to 93 °C. As shown in Fig. 2d, e, f, the IR radiation intensity in the range of 4 μm -20 μm is 248 W m^{-2} , 20.7 W m^{-2} , 16.6 W m^{-2} for the $\text{Bi}_2\text{Te}_3/\text{Cu}$ with 3 μm , 100 nm, 15 nm thicknesses of Bi_2Te_3 thin film, respectively, significantly lower than the corresponding IR radiation of pure Bi_2Te_3 film of 377 W m^{-2} (Supplementary Fig. 1e). Summing up the solar absorptivity and IR emissivity listed in Supplementary Tab.1, the 100 nm thickness of Bi_2Te_3 layer can synergistically absorb 89 % of sunlight and emit 5 % of IR radiation, in other words, this 100 nm thickness of Bi_2Te_3 layer can absorb sunlight energy to the maximum extent and dissipate heat energy to the minimum extent, so that the heat energy converted from sunlight is localized in the interior of the Bi_2Te_3 layer, resulting in high sunlight irradiated temperature.

References

- 4 Chen, Y. L. *et al.* Experimental Realization of a Three-Dimensional Topological Insulator, Bi_2Te_3 . *Science* **325**, 178-181, (2009).
- 5 Dheepa, J., Sathyamoorthy, R. & Subbarayan, A. Optical properties of thermally evaporated Bi_2Te_3 thin films. *J. Cryst. Growth* **274**, 100-105, (2005).
- 15 Lyu, Z. H., Chen, R. H., Mavrikakis, M. & Xia, Y. N. Physical Transformations of Noble-Metal Nanocrystals upon Thermal Activation. *Accounts. Chem. Res.* **54**, 1-10, (2021).
- 16 Zhao, F., Guo, Y. H., Zhou, X. Y., Shi, W. & Yu, G. H. Materials for solar-powered water evaporation. *Nat. Rev. Mater.* **5**, 388-401, (2020).

4. The function of the $\text{Bi}_2\text{Te}_3/\text{Cu}$ interface is to keep the stability of the $\text{Bi}_2\text{Te}_3/\text{Cu}$ heterostructure and to conduct the solar-irradiated high temperature generated by Bi_2Te_3 to the inside of the device for heating the catalyst, which has been described in detail in the manuscript and is also placed below for your convenience.

Supplementary Fig. 3 The lattice fringes of the $\text{Bi}_2\text{Te}_3/\text{Cu}$ film.

“Supplementary Fig. 3 showed the crystal planes of Cu (111), CuO (002), and Bi_2Te_3 (015) grown in the interface. This interface layer of CuO may be caused by the intrinsic oxidation of the Cu substrate in air. Moreover, the CuO interface layer is $\sim 1.5 \text{ nm}$ thick, which is so thin that the sunlight converted heat energy in the Bi_2Te_3 layer can be smoothly transferred to the Cu layer, thus heating the device and catalyst efficiently.”

REVIEWERS' COMMENTS

Reviewer #3 (Remarks to the Author):

The authors well addressed my concern in this version of revised manuscript, I recommend its publication in Nature Commun.

REVIEWER COMMENTS

Reviewer #3 (Remarks to the Author):

The authors well addressed my concern in this version of revised manuscript, I recommend its publication in Nature Commun.

Response: The authors very thank indeed for your professional and kind support.

Through your guidance, we have greatly improved the logic of our manuscript.